# Beyond Model Base Retrieval: Weaving Knowledge to Master Fine-grained Neural Network Design

**Jialiang Wang** [1]  **Hanmo Liu** [1 2]  **Shimin Di** [3]  **Zhili Wang** [1]  **Jiachuan Wang** [4]  **Lei Chen** [2 1]  **Xiaofang Zhou** [1]

## Abstract

Designing high-performance neural networks for new tasks requires balancing optimization quality with search efficiency. Current methods fail to achieve this balance: neural architectural search is computationally expensive, while model retrieval often yields suboptimal static checkpoints. To resolve this dilemma, we model the performance gains induced by fine-grained architectural modifications as edit-effect evidence and build evidence graphs from prior tasks. By constructing a retrieval-augmented model refinement framework, our proposed **M-DESIGN** dynamically weaves historical evidence to discover near-optimal modification paths. M-DESIGN features an adaptive retrieval mechanism that quickly calibrates the evolving transferability of edit-effect evidence from different sources. To handle out-of-distribution shifts, we introduce predictive task planners that extrapolate gains from multi-hop evidence, thereby reducing reliance on an exhaustive repository. Based on our model knowledge base of 67,760 graph neural networks across 22 datasets, extensive experiments demonstrate that M-DESIGN consistently outperforms baselines, achieving the search-space best performance in 26 out of 33 cases under a strict budget.

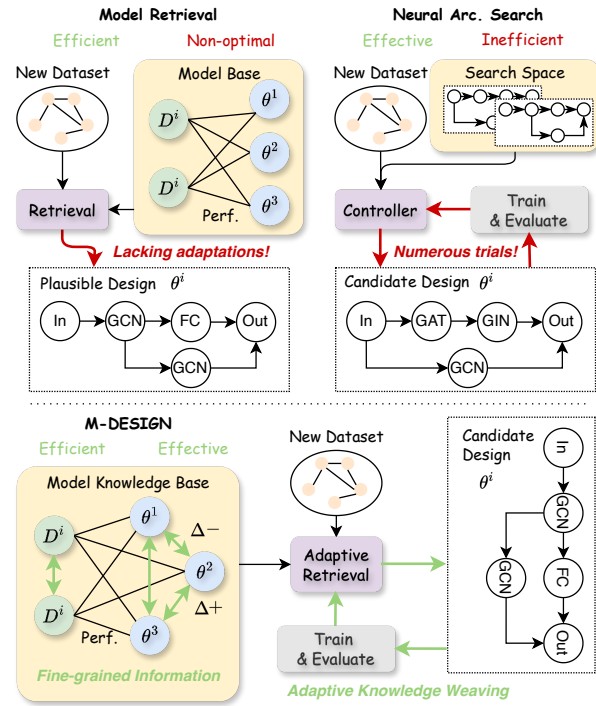

*Figure 1.* Illustration on the effectiveness-efficiency dilemma of conventional neural architecture search and model retrieval methods (Upper part), and the novel framework of M-DESIGN that adaptively weaves fine-grained modification-gain information for a better trade-off (Lower part).

## 1. Introduction

Designing high-performance neural networks is a cornerstone of modern machine learning research and its diverse applications (Trirat et al., 2025; He et al., 2016; Kipf & Welling, 2017; Devlin et al., 2018). However, existing

---

[1]The Hong Kong University of Science and Technology, Hong Kong SAR, China [2]The Hong Kong University of Science and Technology (Guangzhou), Guangzhou, China [3]Southeast University, Nanjing, China [4]University of Tsukuba, Tsukuba, Japan. Correspondence to: Hanmo Liu <hliubm@connect.ust.hk>.

*Proceedings of the 43rd International Conference on Machine Learning*, Seoul, South Korea. PMLR 306, 2026. Copyright 2026 by the author(s).

methodologies often struggle to achieve a satisfactory trade-off between design efficiency and model effectiveness. Classical Neural Architecture Search (NAS) (Zoph et al., 2018; Pham et al., 2018; Zhou et al., 2022; Wang et al., 2025) typically relies on exhaustive trial-and-error to identify superior structures, paying the cost of "reinventing the wheel" before finding a good configuration. To address these inefficiencies, recent studies have explored retrieving models from extensive model repositories and pre-trained model zoos (Wang et al., 2026; Liu et al., 2025a; Ooi et al., 2024; Wu et al., 2023; Li et al., 2024) to bypass searching from scratch. Nevertheless, these retrieval-based approaches rarely deliver an optimal model, due to their extreme focus on setting a reasonable starting point while leaving subsequent model adaptations to ad-hoc trial-and-error. Real tasks, however,

usually require fine-grained architectural edits (e.g., changing message passing in graph neural networks) to match idiosyncratic data properties (e.g., homophilic or heterophilic graphs) and distribution shifts (Wang et al., 2023).

To better balance efficiency and effectiveness, we must move beyond simple model retrieval (Cao et al., 2023; Liu et al., 2025b; Xing et al., 2024; Liu et al., 2025a; Wang et al., 2026; Kumar et al., 2016; You et al., 2020) and focus on identifying the shortest modification trajectory toward the optimum. This requires making **edit-effect evidence** explicit—i.e., structuring how each fine-grained architectural modification affects performance across diverse tasks—rather than storing only coarse performance records of complete models (Qin et al., 2022; Chitty-Venkata et al., 2023). Yet, refining models using edit-effect evidence faces two challenges:

1. **Evolving transferability along refinement:** The transferability of prior evidence is not static during refinement. As the architecture evolves along the modification trajectory, the source tasks that exhibit good edit-effect transferability may change. Task similarity signals computed at the beginning can quickly become miscalibrated.

2. **Evidence breakdown under OOD shift:** Under out-of-distribution (OOD) and limited repository coverage, directly retrieving *1-hop* edit-effect evidence becomes unreliable or even misleading. In such cases, refinement requires predictive reasoning that can extrapolate *multi-hop* modification gains beyond observed local evidence to avoid negative transfer and error accumulation.

To address this gap, we present **M-DESIGN**, a retrieval-augmented framework for mastering neural network refinement by adaptively weaving prior evidence about fine-grained architecture modification. To address **missing edit-effect evidence**, we build a *model knowledge base* (MKB) that organizes curated model records into a *modification-gain graph*, explicitly encoding data properties, architecture variants, and pairwise performance deltas. This representation supports relational reasoning over *what changed* and *how much it helped*, turning model refinement into continuous retrieval over modification gains. To address **evolving transferability**, we propose *knowledge weaving* with a Bayesian belief over task similarity. We use gains observed along the modification trajectory to perform online updates, yielding a *dynamic task similarity* that progressively corrects unreliable priors under distribution shift. To handle **OOD shifts**, when direct evidence retrieval becomes unreliable, M-DESIGN leverages *predictive task planners* trained on the modification-gain graph to estimate *multi-hop* gains and guide exploration under a strict refinement budget.

We instantiate M-DESIGN on graph learning and build an MKB spanning 3 task types, 22 datasets, and 67,760

trained GNN models with structured metadata. Under a unified refinement budget, M-DESIGN consistently outperforms strong retrieval and refinement baselines and discovers the optimal model architecture within our evaluated search space in 26/33 task-data pairs. Beyond aggregate wins, we provide mechanism-level evidence that the proposed dynamic task similarity aligns with refinement outcomes, explaining why M-DESIGN remains reliable when static similarity fails. Our contributions are as follows:

- We reframe model refinement as adaptive retrieval that quickly discovers near-optimal modification paths using repository priors, while dynamically calibrating transfer evidence to reduce reliance on repository completeness.

- We introduce Bayesian dynamic task similarity and an iterative retrieval-and-refine algorithm that updates transfer beliefs online, and trains predictive task planners to extrapolate reliable evidence under OOD shifts.

- We show strong empirical performance on 33 task-data pairs, including 26 search-space-best outcomes under strict budgets, and provide mechanistic analyses explaining the gains. We release a model knowledge base recording 67,760 GNNs.

## 2. Related Works

This work lies at the intersection of automated model design and model retrieval, which are introduced below.

**Automated model design** aims to identify the optimal neural network configuration that maximizes performance (e.g., accuracy or AUC) on a given task and data. Formally, given a task dataset $D$, a candidate model space $\Theta$, and an evaluation function $\mathcal{P}(\theta, D)$ for $\theta \in \Theta$, the goal is to find:

$$\theta^* = \underset{\theta \sim \pi_\alpha(\Theta)}{\operatorname{argmax}} \mathbb{E}[\mathcal{P}(\theta, D)], \qquad (1)$$

where $\pi_\alpha(\Theta)$ is the model sampling strategy. While traditional Neural Architecture Search (NAS) methods (White et al., 2021; Zoph & Le, 2017; Shi et al., 2022; Liu et al., 2019) yield near-optimal solutions, they rely on repeated trials and can fail to converge under limited budgets (Elsken et al., 2019; He et al., 2021; Oloulade et al., 2021). Recent predictor-assisted variants (Sridhar & Chen, 2025) reduce within-task search cost by predicting performance differences between similar architectures. However, most NAS pipelines remain *task-isolated*: the search process for each new task operates independently and does not reuse model-performance records from prior tasks. This cold-start issue underlies their high resource demands, particularly when working with high-dimensional, non-convex spaces.

**Model retrieval** amortizes search costs by reusing historical training records from curated model bases $\mathcal{M}$ and retrieving

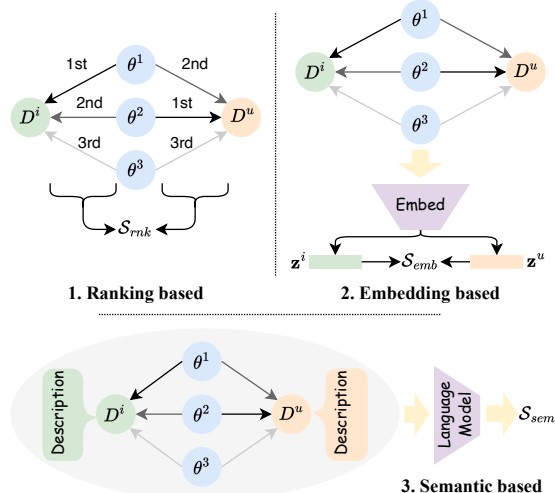

**1. Ranking based**

**2. Embedding based**

**3. Semantic based**

*Figure 2.* Illustration of transferability estimators in model retrieval: 1) ranking-based, 2) embedding-based, and 3) semantic-based.

a promising starting model via *transferability estimation* between a new task $D^u$ and previously benchmarked tasks $\{D^i\}_{i=1}^N$. By noting the transferability estimator as $\mathcal{S}(\cdot, \cdot)$, the retrieval process is formulated as the maximization of the transferability-weighted lookup on prior evaluations:

$$\theta^* = \underset{\theta \in \mathcal{M}}{\arg\max} \, \mathcal{P}(\theta, D^u),$$
$$\mathcal{P}(\theta, D^u) \propto \mathcal{S}(D^u, D^i) \odot \mathcal{P}(\theta, D^i), \qquad (2)$$

where $\odot$ denotes method-specific aggregation between task similarity and historical performance. Current retrieval-based methods differ in the design of transferability estimator, which is summarized below and illustrated in Figure 2:

1. **Ranking-based** methods (You et al., 2020; Qin et al., 2022) estimate task similarity via correlations of performance rankings from a shared set of anchor models $\Theta^a \subset \Theta$ on both $D^u$ and benchmarks: $\mathcal{S}_{\text{rnk}}(D^u, D^i) = \texttt{RankSim}(\mathcal{P}(\Theta^a, D^u), \mathcal{P}(\Theta^a, D^i))$. They then directly select the best benchmark model as the optimum for $D^u$.

2. **Embedding-based** methods (Jeong et al., 2021; Cao et al., 2023; Zhao et al., 2021b; Liu et al., 2025a) learn representations $\mathbf{z}$ of tasks and anchor models from meta-features or learned encoders, and define $\mathcal{S}$ via distances in the latent space: $\mathcal{S}_{\text{emb}}(D^u, D^i) = \texttt{Cosine}(\mathbf{z}^u, \mathbf{z}^i)$.

3. **Semantic-based** methods (Zheng et al., 2023; Zhang et al., 2023; Wang et al., 2025; Trirat et al., 2025; Wang et al., 2026) leverage large language models (LLMs) (Brown et al., 2020; Touvron et al., 2023) to align task descriptions with repository/literature knowledge, producing a semantic notion of task relatedness: $\mathcal{S}_{\text{sem}}(D^u, D^i) = \mathcal{LLM}(D^u, D^i, \Theta^{i*})$.

While Equation (2) is effective for choosing a *starting point*, most existing $\mathcal{S}(\cdot, \cdot)$ are computed once and kept fixed. This static transferability signal is ill-suited to sequential refinement—especially under OOD shift—where the architecture must be iteratively edited to match idiosyncratic data properties and evolving local distributions.

## 3. Preliminaries

Effective model design necessitates adapting transferable knowledge via a sequence of structural modifications rather than static retrieval (Candelieri, 2021). In this section, we formalize how historical modification evidence is structured into an *Architecture Modification-Gain Graph* to support this adaptive process, followed by the definition of the *Adaptive Model Retrieval problem*.

### 3.1. Architecture Modification-Gain Graph

To maximize the efficacy of model design, we focus on the *observed gain* of a candidate modification $\Delta\theta$ at step $t$ for a task $D$, defined as:

$$\Delta P_t(\Delta\theta) \triangleq \mathcal{P}(\theta_t + \Delta\theta, D) - \mathcal{P}(\theta_t, D). \qquad (3)$$

Obtaining this gain via online trial-and-error is computationally prohibitive. However, positing that local modification patterns are transferable, we structure prior design experience for each benchmark task $D^i$ into an **Architecture Modification-Gain Graph**, which is shown in Figure 3:

$$G_\Delta^i = (V, E^i, \omega^i).$$

Here, $V$ denotes the universal set of architecture configurations, and $E^i \subseteq V \times V$ denotes feasible 1-hop modifications. Crucially, each directed edge $e = (\theta, \theta')$ is weighted by its historical performance gain on task $D^i$:

$$\omega^i(e) = \Delta P^i(\theta \to \theta') = \mathcal{P}(\theta', D^i) - \mathcal{P}(\theta, D^i). \qquad (4)$$

This graph formulation renders the stored knowledge *composable*. Unlike isolated model checkpoints, edges represent relative gains that are naturally reversible and can be chained. This allows M-DESIGN to reason about modification trajectories and combine local edit patterns across different tasks to quickly approximate the optimal path for the unseen task.

### 3.2. Problem Formulation

Leveraging the structured knowledge in $G_\Delta$, we now frame the model design process as a retrieval-augmented optimization. The core challenge lies in the fact that the true gain on the unseen task, $\Delta P_t^u(\Delta\theta)$, is unobservable before execution. Furthermore, the optimal modification strategy often varies across different regions of the model-performance landscape due to distribution shifts between tasks.

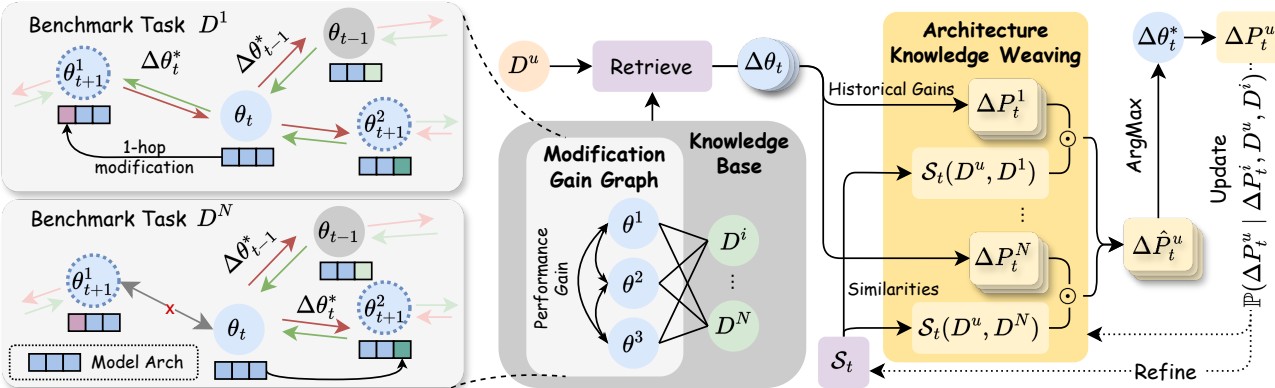

*Figure 3.* Overview of M-DESIGN. The knowledge base stores graph-structured modification evidence (white). A knowledge weaving engine (yellow) aggregates gains from related tasks and updates a Bayesian task-similarity belief online using observation.

Consequently, we define the **Adaptive Model Retrieval Problem** as finding a sequence of modifications that maximizes the expected gain based on historical evidence:

**Definition 3.1** (Adaptive Model Retrieval Problem). Given an unseen task $D^u$ and a knowledge base $\mathcal{K}$ containing modification-gain graphs for known tasks, the refinement process aims to select the optimal architecture modification $\Delta\theta_t^*$ at each step $t$:

$$\Delta\theta_t^* = \underset{\Delta\theta \in \mathcal{C}_t}{\text{argmax}} \, \mathbb{E}[\Delta P_t^u(\Delta\theta) \mid \mathcal{H}_t], \qquad (5)$$

where $\mathcal{H}_t$ is the observed interaction history up to step $t$, and $\mathcal{C}_t$ is the local candidate set retrieved from $\mathcal{K}$ around the current architecture.

This formulation replaces the black-box search controller of traditional NAS with a knowledge-driven retrieval mechanism. The objective implies two requirements: (i) accurately estimating the expected gain $\mathbb{E}[\cdot]$ by aggregating relevant edges from the graph, and (ii) adapting the retrieval strategy online as $\mathcal{H}_t$ accumulates.

## 4. Methodology

This section presents **M-DESIGN**, a retrieval-augmented refinement framework that *weaves fine-grained modification evidence* from prior tasks to guide iterative neural network design on a new task (see Definition 3.1). Our goal is to address two bottlenecks: (i) *evolving edit-effect transferability* along the refinement trajectory, and (ii) *evidence breakdown* under OOD shift and limited repository coverage.

**Overall Workflow.** As illustrated in Figure 3, given an unseen task $D^u$, M-DESIGN iteratively refines an architecture $\theta_t$ for $t = 0, \ldots, T-1$ under a fixed evaluation budget. At each step, we (1) retrieve candidate local modifications $\Delta\theta$ from a model knowledge base $\mathcal{K}$, (2) estimate their expected

gains by *knowledge weaving* across benchmark tasks, (3) execute the best modification on $D^u$ to observe a real gain, and (4) update a *Bayesian belief* over task similarity online so that transfer becomes progressively calibrated.

### 4.1. Fine-grained Knowledge Weaving for Refinement

The major challenge in solving Equation (5) is that the true gain $\Delta P_t^u(\Delta\theta)$ on $D^u$ is *unobservable before execution*, making direct optimization intractable. Our key insight is that tasks with *consistent local modification behavior* tend to share similar refinement landscapes. Therefore, we propose to approximate the gain on $D^u$ using fine-grained gain evidence observed on benchmark tasks $\mathcal{D}^b = \{D^i\}_{i=1}^N$.

**Definition 4.1** (Task Similarity via Local Modification Consistency). Let $D^u$ be an unseen task and $\mathcal{D}^b = \{D^i\}_{i=1}^N$ be benchmark tasks with known evaluations over a model space $\Theta$. For a current architecture $\theta_t \in \Theta$ and a 1-hop modification $\Delta\theta$, we define a task similarity $\mathcal{S}(\cdot, \cdot)$ such that when $\mathcal{S}(D^u, D^i) \geq \delta$, the modification gain on $D^u$ is locally consistent with that on $D^i$:

$$\mathbb{E}[\Delta P_t^u(\Delta\theta) \mid \Delta P_t^i(\Delta\theta), \mathcal{S}(D^u, D^i) \geq \delta]$$
$$= \gamma_{i,t} \, \Delta P_t^i(\Delta\theta) + \epsilon_{i,t}, \qquad (6)$$

where $\Delta P_t^i(\Delta\theta) = P(\theta_t + \Delta\theta, D^i) - P(\theta_t, D^i)$, $\gamma_{i,t}$ captures transfer scaling from $D^i$ to $D^u$, and $\epsilon_{i,t}$ models residual discrepancy.

This definition formalizes a *gain-consistency view* of transferability: high similarity implies that local edits have predictable effects across tasks, which is later validated in Section 5.3. Based on Definition 4.1, we reformulate the refinement objective as *adaptive retrieval by gain weaving*.

**Definition 4.2** (Knowledge Weaving for Adaptive Retrieval). Given an unseen task $D^u$, a current model $\theta_t$, and a model knowledge base $\mathcal{K}$ that stores modification evidence across

benchmark tasks $\mathcal{D}^b = \{D^i\}_{i=1}^N$, the optimal modification $\Delta\theta_t^*$ is selected by maximizing the *woven expected gain*:

$$\Delta\theta_t^* = \underset{\Delta\theta \in \mathcal{C}_t}{\arg\max} \sum_{i=1}^N \mathcal{S}_t(D^u, D^i) \cdot \widetilde{\Delta P}_t^i(\Delta\theta). \quad (7)$$

Here $\mathcal{C}_t$ is the candidate set of feasible local modifications around $\theta_t$, $\mathcal{S}_t(\cdot, \cdot)$ is a *time-varying* task similarity belief, and $\widetilde{\Delta P}_t^i(\Delta\theta)$ denotes the gain evidence used for task $D^i$:

$$\widetilde{\Delta P}_t^i(\Delta\theta) = \begin{cases} \Delta P_t^i(\Delta\theta), \text{if the gain is reliable in } \mathcal{K}, \\ \widehat{\Delta P}_t^i(\Delta\theta), \text{else predicted (see Section 4.3).} \end{cases}$$

Intuitively, Equation (7) generalizes static retrieval. Instead of selecting a whole model based on coarse performance tables, we retrieve *which local edit to apply next* by aggregating fine-grained gain evidence across related tasks.

Under the gain-consistency condition in Definition 4.1, maximizing the woven objective in Equation (7) is a principled surrogate for maximizing the unobserved, expected gain in Definition 3.1. The derivation is shown in Appendix B.1.

## 4.2. Adaptive Retrieval via Dynamic Task Similarity

While the task similarity in Definition 4.1 enables fine-grained knowledge weaving in Definition 4.2 to effectively approximate the model refinement objective in Definition 3.1, we empirically reveal (in Section 5.3) that modification gains can vary drastically across different regions of the landscape, causing global or static task-similarity in Equation (2) to fail.

To make experience transfer *self-correcting*, we maintain a *Bayesian belief* $\mathcal{S}_t(D^u, D^i)$ that is updated online from observed gains on $D^u$. This adaptive schema ensures that if new evidence contradicts the initial guess that $D^i$ is akin to $D^u$, the posterior $\mathcal{S}_t$ gradually downweights $D^i$ to avoid overconfidence.

**Bayesian Similarity Update.** By Bayes' rule, we treat $\mathcal{S}_t^{u,i} = \mathcal{S}_t(D^u, D^i)$ as the updated belief about local similarity and validity of knowledge transfer between $D^u$ and $D^i$, adjusting its probability based on new observations $\Delta P_t^u$:

$$\mathcal{S}_t^{u,i} = \frac{\mathbb{P}(\Delta P_t^u \mid \Delta P_t^i, D^u, D^i) \cdot \mathcal{S}_{t-1}^{u,i}}{\sum_{j=1}^N \mathbb{P}(\Delta P_t^u \mid \Delta P_t^j, D^u, D^j) \cdot \mathcal{S}_{t-1}^{u,j}} \quad (8)$$

where $\mathbb{P}(\Delta P_t^u \mid \Delta P_t^i, D^u, D^i)$ is the likelihood of observing $\Delta P_t^u$ on $D^u$ given $\Delta P_t^i$, and $\mathcal{S}_{t-1}^{u,i}$ is the prior represented by task similarity from the last iteration.

**Likelihood from Gain-consistency.** By Definition 4.1, for a high-similarity benchmark task $D^i$ with $\mathcal{S}(D^u, D^i) \geq \delta$,

we model the relationship between $\Delta P_t^u$ and $\Delta P_t^i$ as a Gaussian distribution with mean $\gamma_{i,t}\Delta P_t^i$ and variance $\sigma^2$. The Gaussian observation model, validated later in Section 5.3, enables smooth probabilistic updates in task similarity, preventing abrupt changes that could lead to instability:

$$\mathbb{P}(\Delta P_t^u \mid \Delta P_t^i, \gamma_{i,t}, \sigma^2) = \mathcal{N}(\Delta P_t^u; \gamma_{i,t}\Delta P_t^i, \sigma^2). \quad (9)$$

where $\gamma_{i,t}$ and $\sigma^2$ are estimated using the history of new observations $(\Delta P^u, \Delta P^i)$ up to iteration $t$, applying sliding windows of size $w$ to control the locality. This Bayesian update has two practical benefits: (i) it avoids overconfident transfer by downweighting tasks that contradict new evidence, and (ii) it enables *online correction* of an imperfect initial similarity prior under out-of-distribution (OOD) shift.

## 4.3. OOD Adaptation with Predictive Task Planners

The retrieval mechanism described above depends on the existence (or completeness) of in-distribution and overlapping edges in $\mathcal{K}$. However, when historical coverage is sparse (missing edges for specific edits) or when $D^u$ is significantly OOD such that all $\mathcal{S}_t(D^u, D^i)$ become negligible, direct retrieval becomes unreliable. To bridge this gap, we augment $\mathcal{K}$ with *Predictive Task Planners* that generate synthetic evidence where historical records are missing or weak.

**Predictive Task Planners.** For each benchmark task $D^i$, we learn a regression model $f_{\psi_i}$ over its modification-gain graph $G_\Delta^i$. This model predicts the gain of a modification $\Delta\theta$ based on the architectural semantics of $\theta_t$:

$$\widehat{\Delta P}_t^i(\Delta\theta) = f_{\psi_i}(\theta_t, \theta_t + \Delta\theta).$$

In practice, $f_{\psi_i}$ is instantiated as an edge-regression GNN (e.g., EdgeConv (Wang et al., 2019)). Crucially, these planners allow us to extrapolate the weaving process to unseen regions of the design space.

**OOD Integration Strategy.** We integrate these predictions into the weaving framework based on the reliability of the knowledge transfer:

- **Filling Missing Edges:** If a potentially optimal modification $\Delta\theta$ has no recorded history in $D^i$ (i.e., edge exists in $G_\Delta^u$ but not $G_\Delta^i$), we substitute the missing value with the predicted gain $\widehat{\Delta P}_t^i(\Delta\theta)$.

- **Weak Transfer Regime:** When the posterior similarity for a task drops below a threshold (i.e., $\mathcal{S}_t(D^u, D^i) \leq \delta$), indicating OOD drift, we rely on the planner's generalized prediction rather than sparse, specific samples.

Furthermore, we maintain a small replay buffer (i.e., representing the interaction history $\mathcal{H}_t$ up to $t$) of observed pairs $((\theta_t, \theta_t + \Delta\theta_t^*), \Delta P_t^u)$ to fine-tune the planners online. This ensures that even in OOD scenarios, the synthetic

*Table 1.* Comparison of search-based and retrieval-based model design methods under a maximum refinement budget of 100 model evaluations across 11 node classification datasets. The best results are **bolded**. * marks reaching global optimum in design space.

| Method | Actor | Computers | Photo | CiteSeer | CS | Cora | Cornell | DBLP | PubMed | Texas | Wisconsin |
|---|---|---|---|---|---|---|---|---|---|---|---|
| **Space Optimum** | **34.89** | **89.59** | **94.75** | **74.59** | **95.33** | **88.50** | **77.48** | **84.29** | **89.08** | **84.68** | **91.33** |
| Random | 33.98 | 88.25 | 94.28 | 73.84 | 94.96 | 87.84 | 74.96 | 83.44 | 88.46 | 80.63 | 89.93 |
| RL | 33.95 | 88.25 | 94.37 | 73.88 | 95.02 | 88.01 | 74.87 | 83.40 | 88.58 | 81.62 | 89.73 |
| EA | 33.73 | 88.28 | 94.45 | 73.93 | 94.94 | 87.90 | 74.24 | 83.66 | 88.42 | 82.43 | 89.20 |
| GraphNAS | 34.11 | 87.94 | 94.38 | 73.89 | 94.90 | 88.01 | 74.60 | 83.40 | 88.58 | 81.98 | 89.87 |
| Auto-GNN | 33.71 | 87.59 | 94.46 | 74.14 | 95.15 | 87.94 | 74.51 | 83.69 | 88.38 | 81.71 | 89.60 |
| GraphGym | 32.72 | 82.57 | 93.53 | 72.68 | 94.71 | 87.95 | 69.37 | 82.54 | 87.61 | 74.77 | 83.33 |
| NAS-Bench-Graph | 32.17 | 82.57 | 93.70 | 65.67 | 93.88 | 88.07 | 69.37 | 83.47 | 88.66 | 74.77 | 86.67 |
| KBG | 32.72 | 87.38 | 94.53 | 74.23 | 94.71 | 87.95 | 69.37 | 83.47 | 88.15 | 74.77 | 86.67 |
| AutoTransfer | 33.97 | 87.72 | 94.62 | 73.89 | 95.16 | **88.50*** | 76.58 | 83.59 | **89.08*** | 78.38 | 88.67 |
| DesiGNN | 34.43 | 88.40 | 94.60 | 74.54 | 95.03 | 88.34 | 75.50 | **84.29*** | **89.08*** | 81.80 | 90.66 |
| **M-DESIGN** | **34.89*** | 89.22 | **94.75*** | **74.59*** | **95.33*** | **88.50*** | **77.48*** | **84.29*** | **89.08*** | 83.79 | **91.33*** |

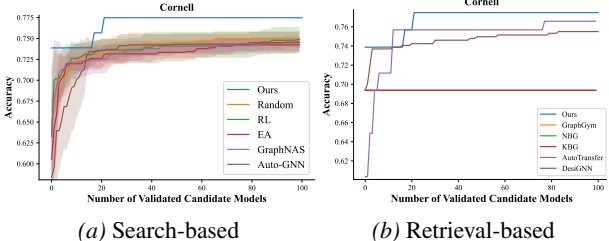

*(a)* Search-based      *(b)* Retrieval-based

*Figure 4.* Long-run model refinement trajectories of M-DESIGN compared to search-based and retrieval-based methods on Cornell.

evidence $\widehat{\Delta P}$ used in Equation (7) gradually aligns with the target distribution, allowing M-DESIGN to transition smoothly from retrieval-based to model-based refinement. In practice, it removes the high-similarity constraint $\delta$ in Definition 4.1 for transferability of evidence (Appendix B.4).

*Table 2.* Comparison of the average number of model evaluations required to reach the high-performance level. Trials that fail to reach the target within 100 evaluations are counted as 100 in the calculation. Lower is better. We mark $\infty$ if a baseline always fails.

| | Ac. | Co. | Ph. | Ci. | CS | C.a | C.l | DB. | Pu. | Te. | Wi. |
|---|---|---|---|---|---|---|---|---|---|---|---|
| Ra. | 86.5 | 71.9 | $\infty$ | $\infty$ | 85.2 | $\infty$ | $\infty$ | $\infty$ | $\infty$ | 74.0 | 79.0 |
| RL | 98.2 | 80.7 | 92.4 | $\infty$ | 67.8 | 94.2 | 92.7 | $\infty$ | $\infty$ | 61.6 | 91.2 |
| EA | 91.4 | 61.7 | 96.4 | $\infty$ | 83.8 | $\infty$ | $\infty$ | 89.1 | $\infty$ | 53.1 | 96.9 |
| Gr. | 95.3 | 92.3 | 87.9 | 82.2 | 89.7 | 97.5 | $\infty$ | $\infty$ | 96.2 | 41.2 | 82.7 |
| AG. | $\infty$ | $\infty$ | $\infty$ | $\infty$ | 52.8 | $\infty$ | $\infty$ | $\infty$ | $\infty$ | 52.0 | 89.7 |
| AT. | $\infty$ | $\infty$ | 65.3 | $\infty$ | 54.6 | **32.8** | $\infty$ | $\infty$ | 79.8 | $\infty$ | $\infty$ |
| De. | 74.0 | 79.4 | 49.2 | 15.4 | 96.8 | $\infty$ | $\infty$ | **3.0** | **3.0** | 40.6 | 62.6 |
| **MD** | **44** | **36** | **21** | **11** | **10** | 46 | **22** | 48 | 50 | **9** | **5** |

## 5. Experiments

Our experiments aim to answer five questions: **(Q1)** Can M-DESIGN reliably find near-optimal architectures under a limited evaluation budget? **(Q2)** Is it more evaluation-efficient than existing search and retrieval-based baselines? **(Q3)** Do the assumptions behind knowledge weaving hold empirically? **(Q4)** Which components are responsible for the gains, especially under OOD settings? **(Q5)** Can M-DESIGN generalize to other data modalities beyond graphs?

### 5.1. Experimental Settings

**Implementation.**[1] We conduct our experiments based on our graph instantiation in Appendix B.4. In total, there are 3 task types, 22 datasets, and 33 task-data pairs (node & link tasks share 11 datasets), all with standard preprocessing (You et al., 2020). This suite spans diverse graph topologies and task difficulties. For each target task-data pair, we

[1] https://github.com/jilwang84/M-DESIGN.

treat the dataset as *unseen* by *removing its performance records from the MKB and never using any test-set signal for retrieval or refinement*. More details are in Appendix C.1.

**Baselines.** We compare with 3 categories of 10 baselines: **(1)** *Search-based Methods*: Random Search (Li & Talwalkar, 2020), Reinforcement Learning (RL) (Zoph & Le, 2017), Evolutionary Algorithms (EA) (Real et al., 2019), Graph-NAS (Gao et al., 2020), and Auto-GNN (Zhou et al., 2022); **(2)** *Retrieval-only Model Selection*: GraphGym (You et al., 2020), NAS-Bench-Graph (Qin et al., 2022), and KBG (Liu et al., 2025a); **(3)** *Retrieval-based Model Refinement*: AutoTransfer (Cao et al., 2023) and DesiGNN (Wang et al., 2026). We equip all the baselines with the *same* design space defined in our model knowledge base for a fair comparison.

**Evaluation Protocol.** Each method is allocated a maximum budget of 100 model evaluations. We report: **(i) Effectiveness:** the best test accuracy / AUC-ROC achieved under the budget; **(ii) Refinement efficiency:** the number of model evaluations required to reach a dataset-specific *high-*

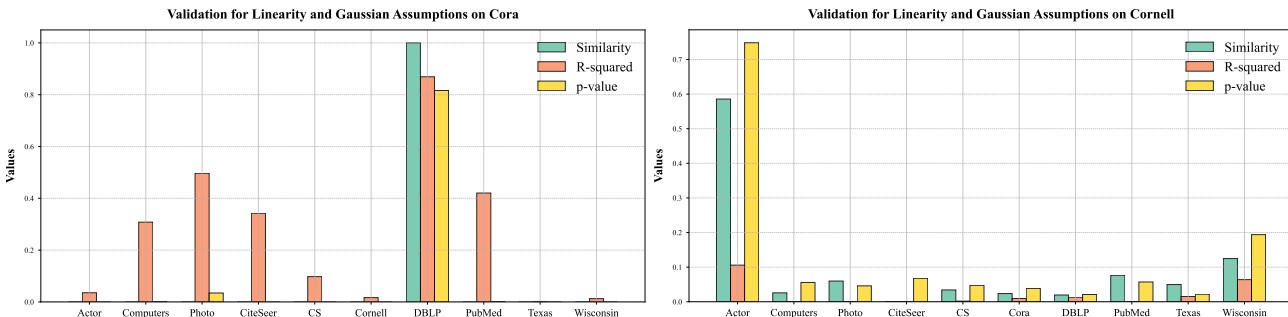

*Figure 5.* Bar charts showing the empirical support for the theoretical assumption on Cora and Cornell. Each benchmark dataset has three bars: our stabilized task similarity snapshot, R-squared (Linearity), and the Shapiro-Wilk p-value (Gaussian).

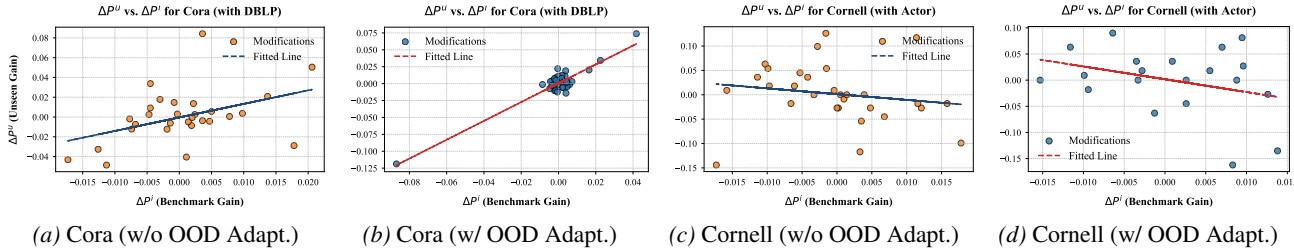

| *(a)* Cora (w/o OOD Adapt.) | *(b)* Cora (w/ OOD Adapt.) | *(c)* Cornell (w/o OOD Adapt.) | *(d)* Cornell (w/ OOD Adapt.) |

*Figure 6.* Scatter plots showing benchmark modification gains (most similar benchmark dataset) vs. unseen modification gains for In-Dist. Cora (left) and Out-of-Dist. Cornell (right). Each dataset has results for two methods: w/ and w/o OOD adaptation.

*performance level*, defined as the best performance attained by M-DESIGN at 50 model evaluations. We report the AUC comparison over best-so-far refinement trajectories. For stochastic baselines, we repeat 10 trials with different seeds.

## 5.2. Main Results: Effectiveness & Efficiency (Q1&Q2)

**Effectiveness (Q1).** Table 1 summarizes node classification results (link prediction and graph classification are reported in Tables 10 and 11). Overall, retrieval-only model selection methods are typically weaker than search-based methods, as they struggle to adapt to unseen tasks. Retrieval-based refinement agents improve over retrieval-only systems but often *prematurely converge* due to static task similarity. M-DESIGN consistently outperforms all baselines and discovers the *best model architecture in the design space* for 26 out of 33 task-data pairs, ranking second-best even on challenging datasets such as Computers and Texas.

**Efficiency (Q2).** In practice, M-DESIGN adds negligible overhead in MKB operations: $< 0.3080$ sec. per iteration (and $< 0.4399$ sec. with OOD adaptation), compared with $\sim 30$ sec. for evaluating one model on Actor using an RTX 3080 GPU. We report the one-time offline MKB construction and storage costs in Appendix B.3. Under our benchmark protocol, in which model evaluation time is replaced by table lookup, we compare *refinement efficiency* by the number of model evaluations required to reach the target performance. As shown in Table 2, many baseline trials fail to reach the preset high-performance level even after 100 evaluations. We further report M-DESIGN's superiority in

the AUC comparison of the best-so-far refinement trajectories in Table 12. Compared with search-based methods, refinement agents are more sample-efficient early on, yet their trajectories in Figures 4 and 9 typically flatten due to local-optimum trapping. In contrast, M-DESIGN maintains strong early outcomes while continuing to improve by dynamically updating task similarity and weaving fine-grained modification knowledge.

## 5.3. Mechanistic Analysis: Assumption Validation (Q3)

The knowledge weaving mechanism in Section 4 builds on three properties of local modification gains: **(A1)** *Linearity*: modification gains transfer approximately linearly between sufficiently similar tasks; **(A2)** *Normality*: modification gains follow a Gaussian distribution, enabling Bayesian updates; **(A3)** *Local consistency*: the true modification consistency varies along refinement trajectories, requiring *dynamic* similarity. We validate them empirically below (**Q3**).

**Linearity and Gaussianity (A1&A2).** Figure 5 reports: (i) M-DESIGN's task similarity snapshot, (ii) $R^2$ for *linearity*, and (iii) Shapiro-Wilk p-values (Shapiro & Wilk, 1965) for *normality*. Based on the overall linearity, we use Cora and Cornell as representatives of in-distribution and OOD cases, respectively. The most similar benchmark datasets (DBLP for Cora; Actor for Cornell) exhibit higher $R^2$ and stronger normality support ($p > 0.05$), validating that *modification gains are more transferable across similar tasks*. This supports the design motivation of knowledge weaving (Definition 4.2) and the probabilistic update in Equation (9).

*Table 3.* Ablation study of M-DESIGN variants and the average Kendall rank correlation with respect to the real local consistency.

| Method | Actor | Computers | Photo | CiteSeer | CS | Cora | Cornell | DBLP | PubMed | Texas | Wisconsin | Avg. Kendall Corr. |
|---|---|---|---|---|---|---|---|---|---|---|---|---|
| **M-DESIGN** | **34.89** | **89.22** | 94.62 | **74.59** | 95.16 | **88.50** | 77.48 | **84.29** | **89.08** | **83.79** | **91.33** | **0.34** |
| w/o windows | **34.89** | **89.22** | 94.60 | **74.59** | 95.06 | **88.50** | 75.68 | **84.29** | 88.66 | 81.98 | **91.33** | 0.27 |
| w/o dynamic | 33.71 | 88.42 | 94.49 | 73.99 | 95.06 | 88.13 | 75.68 | 83.68 | 88.44 | 81.98 | 90.67 | 0.08 |
| w/o OOD | **34.89** | 88.42 | **94.75** | **74.59** | 95.33 | **88.50** | 75.68 | **84.29** | **89.08** | 81.98 | 90.67 | 0.31 |

*Table 4.* Ablation over the MKB scale. We randomly retain a fraction of the benchmark sources and report the final node classification accuracy. Performance degrades gracefully even with only 25% of benchmark tasks retained in the MKB.

| Scale | Actor | Computers | Photo | CiteSeer | CS | Cora | Cornell | DBLP | PubMed | Texas | Wisconsin | Avg. |
|---|---|---|---|---|---|---|---|---|---|---|---|---|
| 25% | 34.12 | 88.49 | 94.62 | 74.19 | 95.06 | 88.32 | 74.78 | 84.00 | **89.08** | 83.79 | 90.00 | 81.50 |
| 50% | **34.89** | 88.45 | 94.62 | **74.59** | 95.21 | 88.26 | 75.68 | **84.29** | 88.70 | 81.08 | **91.33** | 81.55 |
| 75% | 34.28 | 88.49 | 94.62 | 74.19 | 94.89 | 88.38 | **77.48** | **84.29** | **89.08** | 84.68 | 90.00 | 81.85 |
| 100% | **34.89** | **89.22** | **94.75** | **74.59** | 95.33 | **88.50** | **77.48** | **84.29** | **89.08** | 83.79 | **91.33** | **82.11** |

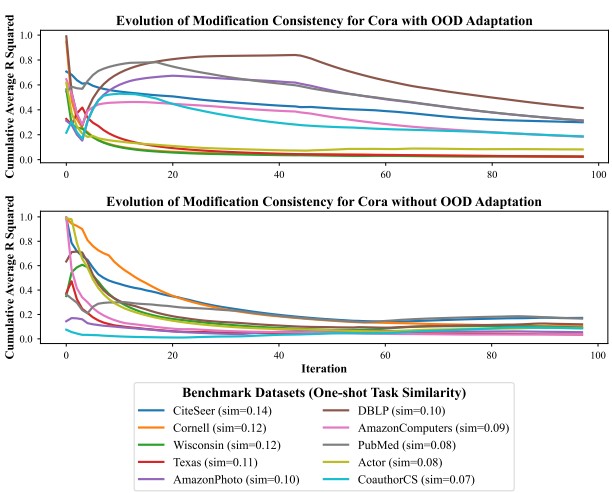

*Figure 7.* Accumulated modification consistency on the Cora dataset with (top) and without OOD adaptation (bottom).

We further visualize the transferability of modification gain in Figure 6. In the in-distribution case, Cora shows a strong linear correlation with DBLP ($R^2 = 0.87$), enabling immediate, reliable reuse of modification knowledge. For Cornell, the correlation is less pronounced without adaptation ($R^2 = 0.03$) but improves after enabling the predictive task planner and OOD adaptation ($R^2 = 0.11$), which explains why OOD-aware updating is crucial for robust refinement.

**Local Consistency (A3).** A central advantage of M-DESIGN is that it quickly *adapts* task similarity using newly observed local modification gains, rather than relying on a static similarity defined at initialization. To demonstrate the necessity of this design, Figure 7 tracks the ground-truth accumulated modification consistency on Cora across refinement iterations. Static similarities listed in the legend often deviate substantially from the true local modification consistency, whereas M-DESIGN's dynamic updates more closely match the evolving refinement landscape. We quantify this behavior in Table 3, where the dynamic similarity variant—

with sliding windows to control the locality—achieves the highest Kendall's $\tau$ rank correlation to the ground truth.

### 5.4. OOD Adaptation & Data Generalization (Q4&Q5)

This section answers **Q4** and **Q5**: which components matter most (especially under OOD), and whether the proposed retrieve-and-refine paradigm transfers beyond graphs. For OOD adaptation studies, we simulate missing edges and weak transfer scenarios, forcing the use of synthetic evidence from predictive task planners.

**Q4: Which components drive the gains?** Table 3 shows that **dynamic similarity update** is the primary driver: removing it (*w/o dynamic*) causes the largest drop in both performance and the alignment between the learned similarity view and the ground-truth local consistency (Avg. Kendall Corr.). **Sliding-window locality** mainly improves robustness by gradually downweighting unreliable evidence from early exploration, while **OOD adaptation** provides the most noticeable benefit on OOD-sensitive datasets (e.g., Computers/Cornell/Texas), mitigating negative transfer when benchmark evidence is sparse or misleading. In practice, we find window sizes around 30–40 offer a robust trade-off between stability and adaptivity across datasets. Table 4 further shows that performance degrades gracefully when only a fraction of benchmark tasks is retained in the MKB.

**Q4: Why does OOD adaptation help?** OOD shift weakens the transferability of modification gains. As illustrated in Figure 6, the gain correlation between the unseen dataset and its closest benchmark can be nearly uninformative without adaptation (e.g., Cornell), which explains why static transfer often fails. Enabling OOD adaptation makes multi-hop gain evidence more predictive and improves refinement stability (as reflected by the increasing consistency trend in Figure 7).

**Q5: Generalization beyond graphs.** We further instantiate M-DESIGN on **tabular** (HPOBench (Klein & Hutter, 2019)) and **image** (NATS-Bench (Dong et al., 2022))

*Table 5.* Performance comparison of model refinement for data-sensitive tabular domain within 100 model evaluations.

| Method | Protein | Slice | Parkinson | Naval |
|---|---|---|---|---|
| **Optimal** | 0.215368 | 0.000144 | 0.004239 | 0.000029 |
| **Random** | 0.260853 | 0.000448 | 0.015955 | 0.000453 |
| **EA** | 0.251967 | 0.000671 | 0.016186 | 0.000110 |
| **GraphG.** | 0.319117 | 0.000799 | 0.018600 | 0.000317 |
| Rank | (8768/62208) | (2150/62208) | (1021/62208) | (1554/62208) |
| **Our-S** | **0.223902** | **0.000221** | **0.012643** | **0.000034** |
| Rank | (**29**/62208) | (**48**/62208) | (**294**/62208) | (**6**/62208) |

*Table 6.* M-DESIGN achieves outstanding results even without refinement in the data-insensitive image domain.

| Method | Cifar10 | Cifar100 | ImageNet16-120 |
|---|---|---|---|
| **Optimal** | 0.8916 | 0.6118 | 0.3827 |
| **M-DESIGN-Init** | 0.8896 | 0.6044 | 0.3827 |
| Model Rank | (**6**/15625) | (**14**/15625) | (**1**/15625) |

benchmarks. The rank denominators in Table 5 and Table 6 denote the full benchmark search spaces: 62,208 candidate models for HPOBench and 15,625 for NATS-Bench. On tabular tasks (Table 5), even a simplified variant (**Our-S**) achieves strong MSE and top model ranks, indicating that refinement remains valuable in *data-sensitive* settings. On images (Table 6), one-shot retrieval (**M-DESIGN-Init**) already reaches near-optimal performance, suggesting that refinement provides smaller marginal gains when data are relatively *homogeneous* and model transfer is easier.

## 6. Limitations and Future Work

Best performance depends on repository coverage and the validity of local gain modeling. It is a local conditional-expectation assumption. While our OOD adaptation can mitigate most distribution shifts and evidence sparsity, extreme OOD tasks or the absence of evidence can reduce reliability and consume more budget. Predictive task planners provide an empirical fallback rather than a theorem-level guarantee. Promising directions include scaling to larger model families, richer edit operators, continually growing knowledge bases, and uncertainty-aware gain prediction.

## 7. Conclusion

We presented **M-DESIGN**, a retrieval-augmented refinement approach that turns historical *edit-effect evidence* into actionable guidance for fine-grained architecture improvement on unseen tasks. M-DESIGN organizes prior evaluations into *modification-gain graphs* and selects the next edit via *knowledge weaving* across related tasks; a Bayesian *dynamic task-similarity* update calibrates transferability online, and *predictive task planners* estimate multi-hop gains

when direct evidence is unreliable. Across 33 task-data pairs under a fixed refinement budget, M-DESIGN outperforms strong baselines and reaches design-space optima in 26/33 cases. We also release a structured knowledge base spanning 3 graph task types, 22 datasets, and 67,760 evaluated models with explicit edit-effect records.

## Acknowledgements

Lei Chen's work is partially supported by National Key Research and Development Program of China Grant No. 2023YFF0725100, National Science Foundation of China (NSFC) under Grant No. U22B2060, Guangdong-Hong Kong Technology Innovation Joint Funding Scheme Project No. 2024A0505040012, the Hong Kong RGC GRF Project 16213620, RIF Project R6020-19, AOE Project AoE/E-603/18, Theme-based project TRS T41-603/20R, CRF Project C2004-21G, Key Areas Special Project of Guangdong Provincial Universities 2024ZDZX1006, Guangdong Province Science and Technology Plan Project 2023A0505030011, Guangzhou municipality big data intelligence key lab, 2023A03J0012, Hong Kong ITC ITF grants MHX/078/21 and PRP/004/22FX, Hong Kong ITC TC-SKLCRCC26EG01, Zhujiang scholar program 2021JC02X170, Microsoft Research Asia Collaborative Research Grant, HKUST-Webank joint research lab, 2025 HKUST Shenzhen-Hong Kong Collaborative Innovation Institute Green Sustainability Special Fund from Shui On Xintiandi and the InnoSpace GBA, and HKUST(GZ) - CMCC(Guangzhou Branch) Metaverse Joint Innovation Lab under Grant No. P00659. Xiaofang Zhou's work is supported by the Hong Kong RGC (grant#C6004-25G, 16210625), HKUST-MetaX Joint Laboratory for Advanced AI Computing (grant#METAX24EG01), and is partially conducted in the JC STEM Lab of Data Science Foundations funded by The Hong Kong Jockey Club Charities Trust. Jiachuan Wang's work is supported in part by JST CREST (JPMJCR22M2). Shimin Di's work is supported by National Science Foundation of China (NSFC) under Grant No. 62506075.

## Impact Statement

This work aims to make neural network architecture refinement more systematic and sample-efficient, thereby reducing ad hoc trial-and-error and the energy costs of exhaustive search. Its risks are indirect: easier refinement may accelerate deployment in sensitive applications, and reused repositories may carry benchmark biases or artifacts. These concerns should be managed through task-appropriate data governance, robustness, and subgroup evaluation, and domain-specific oversight, especially when refined models are used in high-stakes, safety-critical, or fairness-sensitive settings where failures can be consequential.

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

# A. Preliminary

### A.1. Definition of Model Knowledge Base

To bypass the cold-start issue in automated model design, recent studies have treated prior model performance records as a model base $\mathcal{M}$ for retrieval. However, a higher-level model knowledge often refers to structured relations linking task datasets, model architectures, and performance outcomes:

$$\mathcal{K} : \mathcal{D} \times \Theta \to \mathcal{P}, \tag{10}$$

where task datasets $\mathcal{D}$, models $\Theta$, and performance metrics $\mathcal{P}$ are systematically represented as relational schema entries. Acquiring such a model knowledge base $\mathcal{K}$ allows us to predict promising model configurations on new tasks. Currently, such metadata is mainly presented as NAS-style benchmarks (Chitty-Venkata et al., 2023), which record model-performance mappings across diverse benchmark datasets in separate flat tables. However, such an unstructured model *knowledge schema* lacks fine-grained relationality and adaptability to support relational analytics over architecture tweaks, which is important in effective model refinement, motivating our MKB in Appendices B.2 and B.4.

- **Fine-grained Relationality**: As NAS-style benchmarks are developed for efficiently evaluating NAS methods, they lack a fine-grained relational structure that links diverse task specifications with model architecture variations. The transferability of the model knowledge is thus limited.

- **Schema Adaptability**: Existing methods implicitly assume that unseen datasets always distribute similarly to known datasets $D^u \sim \mathcal{D}$. This assumption does not hold in real life, as deviations in topology, feature distributions, or task requirements can easily render unseen tasks OOD. Without mechanisms to detect and adapt to such distributional drift, existing model knowledge bases may provide invalid or harmful knowledge to unseen tasks.

### A.2. Model Base Systems Literature

Early model base systems primarily seek efficiency by systematically collecting, structuring, and retrieving historical model-performance data from model bases (MBs). Pioneering systems (Miao et al., 2016; Vartak et al., 2016) primarily focus on organizing and querying model data to streamline experiment management and facilitate reproducibility. Rafiki (Wang et al., 2018) advances this approach by managing distributed training and hyperparameter tuning workflows via structured model data queries, thereby significantly reducing redundant computational effort. Other systems (Li et al., 2018; Nakandala et al., 2020; Kumar et al., 2016; Wu et al., 2023; Tan et al., 2024) further provide structured, MB-oriented interfaces for efficient model selection, hyperparameter tuning, and resource scheduling, leveraging relational model data management and SQL-style declarative queries. Recently, TRAILS (Xing et al., 2024) introduced a two-phase model selection that integrates training-free and training-based model evaluation within model base systems, thereby efficiently balancing exploration and exploitation while achieving Service-Level Objectives. Collectively, these model base systems emphasize the power of structured model queries to mitigate redundant computations in model selection.

### A.3. Graph Neural Networks Literature

In this paper, we use graph learning as a testbed to evaluate retrieval-augmented model design and refinement methods. Graph data is often represented by a set of nodes and edges $G = \{V, E\}$, where edges $e = (u, v) \in E$ represent relationships between entities $u, v \in V$. To acquire knowledge from graphs, Graph Neural Networks (GNNs) (Veličković et al., 2018; Hamilton et al., 2017; Xu et al., 2019; Defferrard et al., 2016; Bianchi et al., 2022; Morris et al., 2019) have become the dominant learning paradigm, which leverages neighborhood aggregation to encode node representations $\mathbf{H}$. This schema (Gilmer et al., 2017) can be formulated as:

$$\mathbf{H}_v^l = \texttt{TRANS}\big(\mathbf{H}_v^{l-1}, \texttt{AGG}(\{\mathbf{H}_u^{l-1} \mid u \in N(v)\})\big), \tag{11}$$

where $l$ notes the layer index, $N(v)$ is the neighbor set of $v$, $\texttt{AGG}(\cdot)$ aggregates neighbor representations, and $\texttt{TRANS}(\cdot)$ transforms gathered information into next-level representations. The core operation in designing GNNs lies in finding effective $\texttt{AGG}(\cdot)$ and $\texttt{TRANS}(\cdot)$ architectures. This foundational framework has been extensively applied across tasks such as node classification (Kipf & Welling, 2017), link prediction (Zhang & Chen, 2018), and graph classification (Zhang et al., 2018). Graph NAS methods (Wang et al., 2022b; Wei et al., 2021; Wang et al., 2022a; 2021; Gao et al., 2020; Zhao et al., 2021a; Wei et al., 2022) have also been developed to achieve optimal GNN architectures for specific datasets. Related automated-design efforts further search fine-tuning strategies for pre-trained GNNs and scoring/message functions for knowledge graph (Wang et al., 2024; Di & Chen, 2023; Di et al., 2021; 2025).

---

**Algorithm 1** M-DESIGN: Adaptive Knowledge Weaving for Retrieval-augmented Model Refinement

---

**Require:** Unseen task $D^u$, initial model architecture $\theta_0$, initial task similarity $\{\mathcal{S}_0(D^u, D^i)\}_{i=1}^N$, maximum iterations $T$, and OOD threshold $\delta$

**Ensure:** Near-optimal model architecture $\theta^*$ for $D^u$

 1: Initialize $\theta^* \leftarrow \theta_0$, $\theta_t \leftarrow \theta_0$
 2: **for** $t = 0$ **to** $T - 1$ **do**
 3:     **for all** 1-hop candidate modification $\Delta\theta_t \in \mathcal{C}_t$ on $G_\Delta^u(\theta_t)$ **do**
 4:         Compute weaving gain: $\widetilde{\Delta\mathcal{P}}(\theta_t, \Delta\theta_t, D^u) \leftarrow \sum_{i=1}^N \mathcal{S}_t(D^u, D^i) \cdot \widetilde{\Delta P}_t^i(\Delta\theta_t)$
 5:     **end for**
 6:     $\Delta\theta_t^* \leftarrow \arg\max_{\Delta\theta_t \in \mathcal{C}_t} \widetilde{\Delta\mathcal{P}}(\theta_t, \Delta\theta_t, D^u)$
 7:     $\theta_{t+1} \leftarrow \theta_t + \Delta\theta_t^*$
 8:     Evaluate $\theta_{t+1}$ on $D^u$ to obtain $\Delta P_t^u$; replace $\theta^*$ if better
 9:     **for all** benchmark task $D^i \in \mathcal{D}^b$ **do**
10:         Compute likelihood (window size $w$ over past $\gamma$ and $\sigma^2$):
11:         $L_i \leftarrow \frac{1}{\sqrt{2\pi\sigma^2}} \exp\left( - \frac{(\Delta P_t^u - \gamma_{i,t} \Delta P_t^i)^2}{2\sigma^2} \right)$
12:     **end for**
13:     **for all** benchmark task $D^i \in \mathcal{D}^b$ **do**
14:         Update task similarity using Bayes' rule:
15:         $\mathcal{S}_t(D^u, D^i) \leftarrow \frac{L_i \, \mathcal{S}_{t-1}(D^u, D^i)}{\sum_{j=1}^N L_j \, \mathcal{S}_{t-1}(D^u, D^j)}$
16:         **if** $\mathcal{S}_t(D^u, D^i) < \delta$ **then**
17:             Flag $D^i$ as OOD
18:             Update predictive task planner with $\mathcal{H}_t$ for $D^i$
19:         **end if**
20:     **end for**
21:     Add $\big((\theta_t, \Delta\theta_t^*), \Delta P_t^u\big)$ to replay buffer $\mathcal{H}_{t+1}$
22:     $\theta_t \leftarrow \theta_{t+1}$
23: **end for**
24: **return** $\theta^*$

---

# B. Methodology

This appendix provides supplementary methodological details, including the full pseudocode of M-DESIGN (Algorithm 1), the derivation of the woven gain objective (Appendix B.1), the logical schema of the model knowledge base (Appendix B.2), the storage and offline construction cost of the MKB (Appendix B.3), and the MKB for graph data (Appendix B.4).

## B.1. Derivation of the Woven Gain Objective (Equation (7))

Fix iteration $t$ and condition on the current state $(\theta_t, \mathcal{H}_t)$. For each benchmark task $D^i$, let $\Delta P_t^i(\Delta\theta)$ denote the observed gain evidence in $\mathcal{K}$ for any candidate $\Delta\theta \in \mathcal{C}_t$; thus $\Delta P_t^i(\Delta\theta)$ is measurable w.r.t. $\mathcal{H}_t$.

**(A1) Local gain-consistency in conditional expectation.** Definition 4.1 assumes (and is empirically validated in Section 5.3) that for each $i$ with sufficiently high local similarity $\mathcal{S}_t(D^u, D^i) \geq \delta$, the unseen-task gain satisfies:

$$\mathbb{E}\big[\Delta P_t^u(\Delta\theta) \mid \mathcal{H}_t, D^i\big] = \gamma_{i,t} \, \Delta P_t^i(\Delta\theta) + \epsilon_{i,t},$$

where $\gamma_{i,t}$ and $\epsilon_{i,t}$ are estimated from historical observations in $\mathcal{H}_t$.

**(A2) Bayesian model averaging over source-task evidence.** We interpret the dynamic similarity belief in Equation (8) as normalized mixing weights $\{\mathcal{S}(D^u, D^i)\}_{i=1}^N$ (with $\sum_i \mathcal{S}(D^u, D^i) = 1$) that quantify how much each benchmark task's local evidence is trusted at step $t$. Then for any candidate $\Delta\theta$,

$$\mathbb{E}[\Delta P_t^u(\Delta\theta) \mid \mathcal{H}_t] = \sum_{i=1}^N \mathcal{S}(D^u, D^i) \, \mathbb{E}\big[\Delta P_t^u(\Delta\theta) \mid \mathcal{H}_t, D^i\big].$$

*Table 7.* The complete model space in our model knowledge base. Bolded choices have been recorded in the released version.

| Type | Design Dimension | Candidate-set $O$ |
|---|---|---|
| Pre-MPNN | Transformation $\mathbf{f}_\theta$ | $O_{f_\theta} = \{\text{MLP (#layer: 1, }\mathbf{2}, 3)\}$ |
| Intra-layer | Neighbor mechanism $N_k(v)$ | $O_N = \{\mathbf{first\_order}, \mathbf{high\_order}, \mathbf{attr\_rewire}, \mathbf{diff\_rewire}, \mathbf{PE/SE\_rewire}\}$ |
| | Edge weight $e_{uv}^k$ | $O_e = \{\text{non\_norm}, \mathbf{sys\_norm}, \mathbf{rw\_norm}, \mathbf{self\_gating}, \mathbf{rel\_lepe}, \mathbf{rel\_rwpe}\}$ |
| | Intra aggregation $intra\_agg_k$ | $O_{intra\_agg} = \{\mathbf{sum}, \mathbf{mean}, \mathbf{max}, \mathbf{min}\}$ |
| | Activation $\sigma$ | $O_\sigma = \{\text{relu}, \mathbf{prelu}, \text{swish}\}$ |
| | Dropout $r_d$ | $O_{r_d} = \{\mathbf{0}, 0.3, 0.5, 0.6\}$ |
| | Combination $comb_k$ | $O_{comb} = \{\mathbf{sum}, \mathbf{concat}\}$ |
| Inter-layer | Message passing layers $n_l$ | $O_{n_l} = \{2, \mathbf{4}, \mathbf{6}, 8\}$ |
| | Inter aggregation $inter\_agg$ | $O_{inter\_agg} = \{\text{concat, last}, \mathbf{mean}, \mathbf{decay}, \mathbf{gpr}, \mathbf{lstm\_att}, \mathbf{gating}, \mathbf{skip\_sum}, \mathbf{skip\_cat}\}$ |
| Post-MPNN | Transformation $\mathbf{f}_\kappa$ | $O_{f_\kappa} = \{\text{MLP (#layer: 1, }\mathbf{2}, 3)\}$ |
| | Edge Decoding $\mathbf{f}_d$ | $O_{f_d} = \{\mathbf{dot}, \mathbf{cosine\_similarity}, \mathbf{concat}\}$ |
| | Graph Pooling $\mathbf{f}_p$ | $O_{f_p} = \{\mathbf{add}, \mathbf{mean}, \mathbf{max}\}$ |
| Training | Learning rate $r_l$ | $O_{r_l} = \{0.1, \mathbf{0.01}, 0.001\}$ |
| | Optimizer $opt$ | $O_{opt} = \{\text{sgd}, \mathbf{adam}\}$ |
| | Epochs $n_e$ | $O_{n_e} = \{100, 200, \mathbf{300}, 400, 500\}$ |

Combining with (A1) yields:

$$\mathbb{E}[\Delta P_t^u(\Delta\theta) \mid \mathcal{H}_t] = \sum_{i=1}^N \mathcal{S}(D^u, D^i)\,\gamma_{i,t}\,\Delta P_t^i(\Delta\theta) + \sum_{i=1}^N \mathcal{S}(D^u, D^i)\epsilon_{i,t}.$$

Since $\gamma_{i,t}$ and $\epsilon_{i,t}$ do not depend on $\Delta\theta$ within the 1-hop neighborhood $\mathcal{C}_t$, we can absorb it into a per-task calibrated evidence $\widetilde{\Delta P}_t^i(\Delta\theta) \triangleq \gamma_{i,t}\Delta P_t^i(\Delta\theta) + \epsilon_{i,t}$, yielding Equation (7).

### B.2. Data Model and Logical Schema

In our model knowledge base, we organize all benchmark records we collected as modification-gain graphs, and construct three core relations linked by foreign keys (`task_id`, `arch_id`). These tables form our logical schema for both static lookup and adaptive refinement:

- **Tasks:** (`task_id`, `dataset_id`, `feature_vector`, `task_type`, ...)—stores each task's ID and its precomputed statistical data properties (e.g. homophily, density).

- **Architectures:** (`arch_id`, `design_tuple`, ...)—an entry per unique model architecture, where design choices and hyperparameters are encoded in `design_tuple`.

- **Modification Gains:** (`task_id`, `arch_from`, `arch_to`, `gain`)—a pairwise relation capturing every 1-hop architecture tweak (`arch_from`→`arch_to`) with the performance gain on the task.

### B.3. Storage and Offline Construction Cost

M-DESIGN does not require loading or serving all historical model weights during online refinement. Its reasoning operates on the structured schema above, which stores tasks, architecture descriptors, and modification gains. The released MKB occupies about 47.15 MB for 67,760 evaluated model configurations across 33 task-dataset databases. Each task-specific modification-gain graph occupies about 1178 KB, and each pretrained predictive task planner occupies about 112 KB. The one-time offline population cost depends on the dataset and hardware: on our RTX 3080, the longest graph benchmark collection was Physics at about 37.59 GPU-hours, while the fastest finished within 1 GPU-hour. Full graph population is not mandatory because missing edges can be approximated by predictive task planners, as discussed in Section 4.3.

### B.4. Model Knowledge Base for Analyzing Graph

Graph data is a notoriously data-sensitive domain for automated model design because the topological diversity in graph data has a larger impact on architecture performance than in other data modalities (You et al., 2020). As discussed in Appendix A.3, there is no universal GNN architecture that can effectively handle all graph data. Thus, we instantiate M-DESIGN for GNN refinement and construct our MKB, enriching the existing benchmarks with 3 task types and 22 graph

*Table 8.* Benchmark task datasets statistics in MKB.

| Dataset | #Graph | #Node | #Edge | #Feature | #Class | Task |
|---|---|---|---|---|---|---|
| Actor | 1 | 7,600 | 30,019 | 932 | 5 | Node/Link |
| Computers | 1 | 13,752 | 491,722 | 767 | 10 | Node/Link |
| Photo | 1 | 7,650 | 238,162 | 745 | 8 | Node/Link |
| CiteSeer | 1 | 3,312 | 4,715 | 3,703 | 6 | Node/Link |
| CS | 1 | 18,333 | 163,788 | 6,805 | 15 | Node/Link |
| Cora | 1 | 2,708 | 5,429 | 1,433 | 7 | Node/Link |
| Cornell | 1 | 183 | 298 | 1,703 | 5 | Node/Link |
| DBLP | 1 | 17,716 | 105,734 | 1,639 | 4 | Node/Link |
| PubMed | 1 | 19,717 | 88,648 | 500 | 3 | Node/Link |
| Texas | 1 | 183 | 325 | 1,703 | 5 | Node/Link |
| Wisconsin | 1 | 251 | 515 | 1,703 | 5 | Node/Link |
| COX2 | 467 | 41.2 | 43.5 | 35 | 2 | Graph |
| DD | 1,178 | 284.3 | 715.7 | 89 | 2 | Graph |
| IMDB-B. | 1,000 | 19.8 | 96.5 | 136 | 2 | Graph |
| IMDB-M. | 1,500 | 13.0 | 65.9 | 89 | 3 | Graph |
| NCI1 | 4,110 | 29.9 | 32.3 | 37 | 2 | Graph |
| NCI109 | 4,127 | 29.7 | 32.1 | 38 | 2 | Graph |
| PROTEINS | 1,113 | 39.1 | 72.8 | 3 | 2 | Graph |
| PTC_FM | 349 | 14.1 | 14.5 | 18 | 2 | Graph |
| PTC_FR | 351 | 14.6 | 15.0 | 19 | 2 | Graph |
| PTC_MM | 336 | 14.0 | 14.3 | 20 | 2 | Graph |
| PTC_MR | 344 | 14.3 | 14.7 | 18 | 2 | Graph |

*Table 9.* Comparison of the optimal model in our MKB with NAS-Bench-Graph (NBG) on overlapping datasets.

| Method | Cora | CiteSeer | PubMed | CS | Computers | Photo |
|---|---|---|---|---|---|---|
| **NBG (Chitty-Venkata et al., 2023)** | 86.22 | 74.03 | 88.12 | 91.26 | 85.58 | 92.78 |
| **M-DESIGN** | **88.50** | **74.59** | **89.08** | **95.33** | **89.59** | **94.75** |

datasets, and contributing 67,760 additional models along with their performance records. The model design space in our MKB is presented in Table 7, which follows the well-studied message-passing schema (You et al., 2020; Wang et al., 2023).

1. *Broader Graph Tasks and Detailed Topology:* We pioneer an extension beyond the node classification task in graph learning to include both link prediction and graph classification tasks—covering 11 representative datasets for each task (see Table 8)—and explicitly document rich graph topology that may affect model selection (Wang et al., 2023; 2026).

2. *Specialized Model Space:* We focus on fine-grained architecture dimensions (6,160 unique models from Table 7) grounded in message-passing schema (You et al., 2020; Wang et al., 2023), benchmarking higher performance ceiling compared to existing NAS benchmarks (Qin et al., 2022) (shown in Table 9).

3. *Data-driven Architecture Insights:* Unlike existing NAS benchmarks that store raw performance records without explainability, our MKB implicitly encodes correlations between graph topology (e.g., homophily ratio), GNN architecture choices (e.g., aggregation), and expected performance impacts.

In our pipeline, M-DESIGN can initialize task similarity $\mathcal{S}_0(D^u, D^i)$ quickly before any direct model testing on $D^u$ by directly analyzing these underlying correlations using large language models (Wang et al., 2026) or statistical methods like 1) average Kendall's $\tau$ rank correlation over data statistic or 2) Principal Component Analysis (PCA) (Pearson, 1901). To obtain an initial model, we take the weighted average of the best-performing models' architecture choices across each benchmark dataset, with weights determined by the initial task similarity. From our empirical study in Figure 8, it achieves better results than traditional PCA (Pearson, 1901). The high-similarity threshold $\delta$ is discarded for better empirical performance with predictive task planners. By combining these, our model knowledge base is not simply a static collection of $(\theta, \mathcal{P}(\theta, D^i))$ pairs; rather, it is a structured knowledge schema of modification-gain graphs that enables more intelligent utilization of prior model design knowledge.

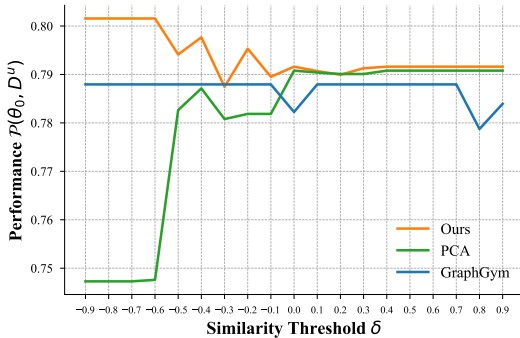

*Figure 8.* Averaged performance of M-DESIGN under different task similarity initialization strategies and initial similarity cutoff $\delta_0$.

*Table 10.* Comparison of search-based and retrieval-based model design methods under a maximum refinement budget of 100 model evaluations across 11 link prediction datasets. The best results are **bolded**. * marks reaching global optimum in design space.

| Method | Actor | Computers | Photo | CiteSeer | CS | Cora | Cornell | DBLP | PubMed | Texas | Wisconsin |
|---|---|---|---|---|---|---|---|---|---|---|---|
| **Space Optimum** | **74.75** | **93.61** | **94.44** | **77.66** | **90.26** | **78.56** | **76.77** | **86.68** | **85.55** | **73.15** | **71.77** |
| Random | 73.76 | 93.26 | 93.94 | 74.24 | 89.38 | 76.66 | 73.84 | 85.48 | 83.13 | 68.70 | 68.98 |
| RL | 73.92 | 93.19 | 94.06 | 74.24 | 89.26 | 76.40 | 73.38 | 85.37 | 83.60 | 69.07 | 68.30 |
| EA | 73.88 | 92.89 | 94.17 | 74.00 | 89.30 | 75.22 | 74.60 | 85.54 | 83.76 | 68.52 | 68.68 |
| GraphNAS | 73.97 | 93.04 | 93.99 | 74.18 | 89.48 | 76.30 | 74.40 | 85.75 | 83.87 | 69.95 | 68.47 |
| Auto-GNN | 74.58 | 93.34 | 94.08 | 74.68 | 89.46 | 74.09 | 75.36 | 85.10 | 82.86 | 70.18 | 69.93 |
| GraphGym | 72.35 | 86.70 | 93.00 | 71.91 | 87.63 | 70.88 | 69.19 | 85.00 | 74.88 | 66.67 | 66.33 |
| NAS-Bench-Graph | 70.33 | 92.94 | 93.00 | 71.91 | 87.63 | 70.88 | 64.14 | 84.46 | 82.54 | 66.67 | 66.67 |
| AutoTransfer | 74.65 | **93.61*** | 93.98 | 73.35 | 90.13 | 75.83 | 74.75 | 85.75 | 84.86 | **73.15*** | 70.07 |
| DesiGNN | **74.75*** | **93.61*** | 94.33 | **75.11** | **90.26*** | 75.67 | **76.77*** | 86.56 | **85.55*** | 71.29 | **71.77*** |
| **M-DESIGN** | **74.75*** | **93.61*** | **94.44*** | 74.98 | **90.26*** | **78.56*** | **76.77*** | **86.68*** | **85.55*** | **73.15*** | **71.77*** |

## C. Experiments

This section summarizes supplementary experimental results that complement the main text: (i) full per-dataset performance tables for link prediction and graph classification under the unified refinement budget, (ii) Area Under the Curve (AUC) results for the refinement trajectories, (iii) analyses of the code-start stability, and (iv) long-run refinement trajectories illustrating how M-DESIGN improves architectures over iterations compared to search-based and retrieval-based baselines.

### C.1. Implementation Details

M-DESIGN is implemented in PyTorch (Paszke et al., 2019) and LangChain (Chase, 2022) (GPT-4o), running on a single RTX 3080 GPU. Our data collection pipeline for the released MKB is built upon the GraphGym platform (You et al., 2020). The modification-gain graphs are structured following the data standard in PyG (Fey & Lenssen, 2019). We implemented GNN-based predictive task planners with two layers and a hidden dimension of 64 using PyG's EdgeConv. For planner pretraining, we use a 5% validation split, Adam with a learning rate of 0.01, and early stopping with patience 20. Online adaptation stores recent unseen-task observations in a replay buffer of size 10, updated in FIFO order, and incrementally fine-tunes the predictive planners by blending buffer samples with historical benchmark edges. To prevent data leakage from the MKB, we excluded the unseen dataset from the benchmark list during evaluation. All results are averaged over 10 repeated trials.

### C.2. Full Per-dataset Results

Tables 10 and 11 report complete results across all datasets for link prediction and graph classification. M-DESIGN consistently matches or outperforms all baselines on every dataset, demonstrating its effectiveness and robustness across diverse graph tasks and data distributions. Overall, it achieves the optimal refinement outcome within our evaluated search space in 26/33 cases.

*Table 11.* Comparison of search-based and retrieval-based model design methods under a maximum refinement budget of 100 model evaluations across 11 graph classification datasets. The best results are **bolded**. * marks reaching global optimum in design space.

| Method | COX2 | DD | IMDB-BINARY | IMDB-MULTI | NCI1 | NCI109 | PROTEINS | PTC_FM | PTC_FR | PTC_MM | PTC_MR |
|---|---|---|---|---|---|---|---|---|---|---|---|
| **Space Optimum** | **86.74** | **80.43** | **81.00** | **48.56** | **80.82** | **78.83** | **79.73** | **70.53** | **68.57** | **73.63** | **66.67** |
| Random | 85.24 | 79.86 | 79.98 | 47.33 | 78.82 | 77.47 | 78.53 | 69.04 | 66.71 | 69.90 | 62.11 |
| RL | 84.80 | 79.22 | 80.08 | 47.65 | 78.27 | 76.71 | 78.92 | 69.18 | 66.10 | 70.55 | 61.52 |
| EA | 84.77 | 79.16 | 79.28 | 47.29 | 78.33 | 77.49 | 78.67 | 68.75 | 65.76 | 69.06 | 61.13 |
| GraphNAS | 85.20 | 79.57 | 80.00 | 47.32 | 78.26 | 76.75 | 78.83 | 68.70 | 66.47 | 70.20 | 61.42 |
| Auto-GNN | 85.41 | 79.69 | 79.12 | 46.97 | 77.73 | 76.24 | 77.69 | 68.12 | 65.05 | 70.50 | 59.36 |
| GraphGym | 72.76 | 73.90 | 72.17 | 46.56 | 74.49 | 69.25 | 76.13 | 65.22 | 59.05 | 59.20 | 50.49 |
| NAS-Bench-Graph | 82.44 | 62.84 | 75.83 | 44.67 | 74.49 | 69.25 | 77.63 | 47.34 | 63.34 | 58.70 | 54.90 |
| AutoTransfer | **86.74*** | 78.44 | 80.17 | 47.22 | 79.00 | 77.74 | 78.38 | 67.15 | **67.14** | 70.15 | 61.76 |
| DesiGNN | 84.95 | 79.04 | 80.73 | 47.44 | 78.95 | 77.58 | **79.73*** | 69.09 | **67.14** | 70.55 | 62.25 |
| **M-DESIGN** | **86.74*** | 80.14 | **81.00*** | 47.65 | 79.04 | **78.83*** | **79.73*** | **70.53*** | 67.14 | **73.63*** | **66.67*** |

*Table 12.* AUC of best-so-far refinement trajectories on node classification. Higher is more efficient.

| Method | Actor | Computers | Photo | CiteSeer | CS | Cora | Cornell | DBLP | PubMed | Texas | Wisconsin | Avg. |
|---|---|---|---|---|---|---|---|---|---|---|---|---|
| Random | 0.338 | 0.872 | 0.940 | 0.734 | 0.948 | 0.867 | 0.739 | 0.825 | 0.882 | 0.792 | 0.890 | 0.802 |
| RL | 0.336 | 0.873 | 0.941 | 0.736 | 0.948 | 0.874 | 0.738 | 0.825 | 0.884 | 0.800 | 0.888 | 0.804 |
| EA | 0.334 | 0.876 | 0.940 | 0.735 | 0.948 | 0.872 | 0.731 | 0.824 | 0.883 | 0.804 | 0.885 | 0.803 |
| GraphNAS | 0.337 | 0.871 | 0.941 | 0.734 | 0.948 | 0.872 | 0.735 | 0.828 | 0.883 | 0.807 | 0.888 | 0.804 |
| Auto-GNN | 0.334 | 0.873 | 0.944 | 0.741 | 0.950 | 0.877 | 0.732 | 0.830 | 0.882 | 0.793 | 0.879 | 0.803 |
| AutoTransfer | 0.335 | 0.875 | 0.945 | 0.738 | 0.949 | **0.883** | 0.750 | 0.833 | 0.887 | 0.780 | 0.880 | 0.805 |
| DesiGNN | 0.341 | 0.877 | 0.945 | 0.741 | 0.949 | 0.882 | 0.747 | **0.843** | **0.891** | **0.809** | 0.898 | 0.811 |
| **M-DESIGN** | **0.342** | **0.885** | **0.946** | **0.743** | **0.951** | **0.883** | **0.768** | 0.839 | 0.887 | **0.809** | **0.907** | **0.815** |

*Table 13.* Initialization sensitivity on node classification. Final accuracies remain close across initialization strategies, indicating that M-DESIGN is robust to cold-start choices.

| Init Strategy | Actor | Computers | Photo | CiteSeer | CS | Cora | Cornell | DBLP | PubMed | Texas | Wisconsin | Avg. |
|---|---|---|---|---|---|---|---|---|---|---|---|---|
| Weighted Avg. | 34.15 | **89.22** | **94.75** | 74.54 | **95.33** | **88.50** | **77.48** | **84.29** | **89.08** | 81.08 | **91.33** | **81.80** |
| Majority Vote | **34.89** | 88.40 | 94.62 | **74.59** | 95.06 | 88.34 | 75.50 | 84.15 | 89.00 | **83.79** | **91.33** | 81.79 |
| Most Similar | 33.88 | 88.64 | 94.48 | 74.28 | 94.90 | 88.26 | 74.57 | 84.04 | 88.96 | 81.08 | **91.33** | 81.31 |

## C.3. AUC Refinement Efficiency

Table 12 reports the area under the best-so-far refinement trajectory for node classification, complementing the target-threshold efficiency metric in the main text. Higher AUC indicates better anytime refinement performance under the same evaluation budget. M-DESIGN obtains the best average AUC while remaining competitive on every dataset.

## C.4. Cold-start Stability

M-DESIGN stabilizes early refinement in three ways. First, it initializes $\mathcal{S}_0(D^u, D^i)$ from explicit data statistics, including graph topology measures such as degree, node count, density, homophily ratio, clustering coefficient, betweenness centrality, and feature dimension. Second, it keeps the prior regression parameters until at least three unseen-task gain pairs are observed, using $\gamma_{i,0} = 1.0$ and $\sigma_0^2 = 0.01$ before that point. Third, the normalized Bayesian update in Equation (8) changes beliefs smoothly rather than making hard source-task switches. Table 13 shows that final performance is stable under three initialization strategies, suggesting that initialization mainly affects the starting point rather than the final refined architecture.

## C.5. Long-run Refinement Trajectories

Figure 9 visualizes long-run refinement behaviors on the remaining 10 node classification datasets, complementing the main text's Figure 4. The shaded regions represent standard deviations over 10 trials. Since most retrieval-based methods lack stochasticity, their curves are solid lines without shading. Compared to search-based methods (which often spend many iterations exploring low-gain edits) and retrieval-based methods (which may plateau after warm-start selection), M-DESIGN more consistently accumulates gains by retrieving transferable 1-hop evidence and, when necessary, leveraging multi-hop gain predictions.

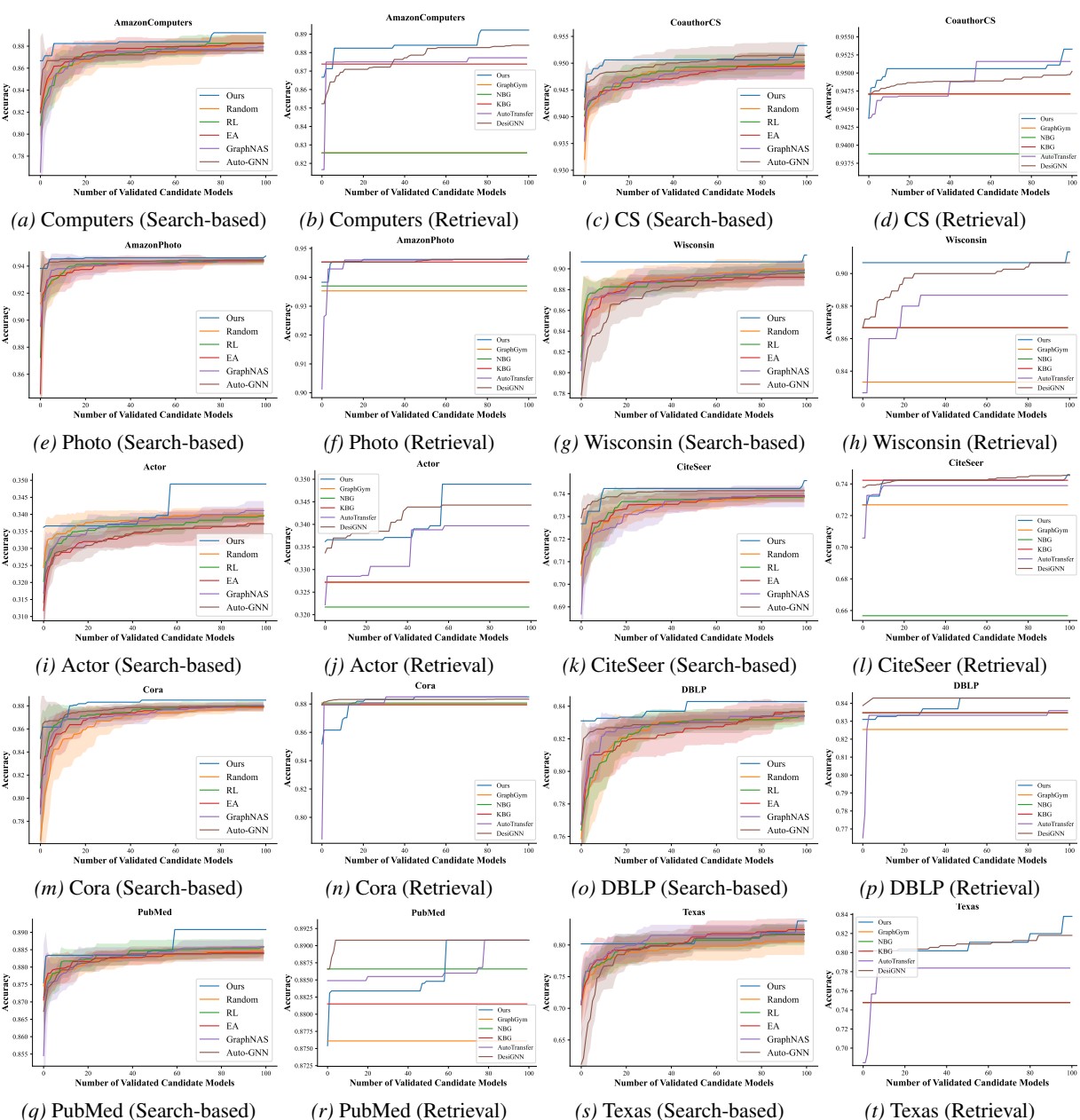

*Figure 9.* Long-run model refinement trajectories of M-DESIGN compared to Search-based and Retrieval-based methods.

