# OpenReview forum: "Beyond Model Base Retrieval: Weaving Knowledge to Master Fine-grained Neural Network Design"
_ICML.cc/2026/Conference — ICML 2026 regular_

### Official Review · Reviewer_1Txj · 2026-02-24

**Soundness:** 3
**Presentation:** 3
**Significance:** 3
**Originality:** 3
**Overall Recommendation:** 4
**Confidence:** 2

**Summary:**

Neural network Architecture Search (NAS) is usually computationally expensive due to its large search space and the expense of evaluating every candidate nerual network. This paper proposes M-DESIGN, a retrieval-augmented framework for efficiently designing high-performance neural networks for new tasks. The authors argue that existing approaches face a trade-off: NAS is effective but computationally expensive, while model retrieval is efficient but often suboptimal because it returns static models without sufficient adaptation. The paper introduces the idea of modeling fine-grained architectural modifications as reusable “edit-effect” evidence. Instead of storing only whole-model performance, the authors construct an architecture Modification-Gain graph, which records how specific small architectural changes affect performance across prior tasks. Experiments on 33 task-dataset pairs show that M-DESIGN outperforms search-based and retrieval-based baselines under strict evaluation budgets, achieving the best architecture in the search space in 26 out of 33 cases.

**Compliance With Llm Reviewing Policy:**

Affirmed.

**Final Justification:**

I keep my weak accept recommendation.

**Key Questions For Authors:**

Q1: Does equaltion 6  in Page 4 always hold? And how to get \gamma_{i,t}? \
Q2: For OOD integration strategy, why there are missing edges? Can we pre-compute all the potential neural network chages and put them into the graph? \
Q3: What guarantees that the planner performs well in the weak-transfer regime? \
Q4: Can you provide an example of OOD adaptation to illustrate how the planner adapts? \
Q5: What does 62208 mean in Table 4 and 15625 mean in Table 5? \
Q6: Can you compare this work with https://ieeexplore.ieee.org/abstract/document/10943463?

**Limitations:**

yes

**Strengths And Weaknesses:**

Strength:
1. The paper proposes M-DESIGN, a retrieval-augmented framework for efficiently designing high-performance neural networks for new tasks. The method estimates expected performance gains of the proposed neural network on new tasks by weaving together evidence from related prior tasks and performs well on graph neural network archetecture search.
2. The paper proposes a dynamic task similarity method to self-correct the task matching. which shown to improve the NAS performance.
3. The proposed method can be generalized to CV and tabular domain.

Weakness:
1. The key insight that tasks with consistent local modification behavior tend to share similar refinement landscapes does not have explicit envidence.
2. How to solve the cold start issue of task matching (new task vs. prior tasks) is not discussed in the paper.
3. The evaluation results do not report error variances.

---

> ### Author Rebuttal · Authors · 2026-03-30
>
> We sincerely thank you for recognizing M-DESIGN's strong performance, the value of dynamic task similarity, and its potential beyond graphs. Below we clarify the points that were not yet explicit enough in the paper.
>
> **W1: Evidence for similar refinement landscapes.**
>
> We presented the evidence in **Sec. 5.3**:
>
> - **Fig. 5, 6, and 7** visualize **modification consistency** and its evolution during refinement. A higher $R^{2}$ is direct empirical evidence for similar local refinement landscapes.
> - We study a strong in-distribution case (Cora-DBLP, $R^{2}=0.87$) and a weak-transfer & OOD case (Cornell, $R^{2}=$ 0.03 without adaptation and 0.11 with OOD adaptation).
> - The higher **Avg. Kendall Corr. in Tab. 3** explains that dynamic similarity tracks the real refinement landscape better than static similarity.
>
> **W2: Cold-start issue.**
>
> This is handled by the **initial similarity prior $S_0(D_u,D_i)$ and the corresponding initialization procedure in Appx. B.3**. Before any unseen-task evaluations, M-DESIGN initializes task similarity by comparing data statistics (e.g., density for graphs) and then constructs an initial model through a similarity-weighted selection of the best-performing architecture choices on each benchmark. **Fig. 8 shows this handle cold-start issue better than other methods.** Please also refer to **D48Y-Q1** for a new sensitivity study on initialization.
>
> **W3: Report error variances.**
>
> **Fig. 9 in Appx. C.1** illustrates the error variances of different methods with shadow areas. **We also add the error variance in the Table of Reply to BwDF-W4** (the 100% row is M-DESIGN in the main table). We will add those details to the experiments in our paper.
>
> **Q1: Does Eq. 6 always hold? How to get $\gamma_{i,t}$?**
>
> No, Eq. (6) is **not claimed to always hold globally**. It is a **local conditional-expectation assumption** used when benchmark-task evidence is sufficiently similar to the unseen task during refinement. Appx. B.1 also specifies that **$\gamma_{i,t}$ and $\epsilon_{i,t}$ are estimated from historical observations in $H_t$**. Eq. (9) further clarifies that the Gaussian update uses the history of ($\Delta P_u$, $\Delta P_i$) pairs with a sliding window to control locality.
>
> **Q2: Can we pre-compute all potential edges?**
>
> Our released MKB has pre-computed all potential model changes and put them into the graph. However, in practice—and following continuous development—the newly constructed graphs may not always be complete, causing direct retrieval to fail. Therefore, as explained in **Sec. 4.3**, we design the predictive task planner also as a backup solver. In case an edge is missing, the predictive task planner could use its neighborhood information to effectively predict the missing modification gains.
>
> **Q3: What guarantees that the planner performs well when OOD?**
>
> We do **NOT** claim a theorem-level guarantee here. Our claim is empirical in Sec. 5.3: when static evidence becomes unreliable, the predictive task planner provides a smoother fallback than continuing to trust sparse or misleading direct reuse. [1] provide a principled expressive foundation for using GNNs as function approximators over graph-structured data, making them a natural choice for predicting local and multi-hop edit effects on the modification-gain graph.
>
> [1] Xu et al. 2018. How powerful are graph neural networks?
>
> **Q4: An example of OOD adaptation.**
>
> The Cornell case in Sec. 5.3 and 5.4 is exactly such a weak-transfer & OOD stress example (i.e., the lowest $R^{2}$). Fig. 5, 6, 7 show why Cornell is OOD and how OOD adaptation works.
>
> **Q5: How to interpret Tab. 4 and 5?**
>
> These are the **total number of candidate models in the corresponding benchmark search spaces** used in the non-graph experiments. The image-table snippet in the paper, e.g., reports model ranks such as (1/15625) (meaning top 1 among 15625 possible models). We will add a clearer explanation.
>
> **Q6: Compare with Delta-NAS?**
>
> We will cite this related work. The key distinction is that **Delta-NAS improves local search *within a task's search space***, while **M-DESIGN improves *cross-task reuse and refinement*** by making historical edit-effect evidence explicit and adaptive.
>
> - **Delta-NAS** is a **task-internal predictor-based NAS** that predicts the difference in accuracy between pairs of similar architectures, projects the architecture space into a lower-dimensional difference space, and then performs an evolutionary search.
> - In contrast, **M-DESIGN** is a **cross-task retrieval-augmented refinement** framework: it stores historical modification gains across tasks, dynamically reweights benchmark tasks through Bayesian task similarity, and invokes predictive task planners when direct evidence is missing or unreliable under OOD shift.
> - Delta-NAS initializes search from a **random population** in a search space, whereas M-DESIGN starts from **cross-task similarity priors** and refines them adaptively.

---

> > ### Author Rebuttal · Reviewer_1Txj · 2026-04-01
> >
> > I think the authors answered all my questions. I would suggest authors can include the answers to Q1, Q3 in the main paper as limitations of the paper.
> > I would like to keep my score.

---

> > > ### Author Response · Authors · 2026-04-01
> > >
> > > Thank you very much for your valuable feedback and for confirming that our rebuttal addressed your concerns. We really appreciate your suggestion to make the points in Q1 and Q3 as the limitation discussion in the main paper, and we will incorporate that in the paper.

---

### Official Review · Reviewer_BwDF · 2026-03-05

**Soundness:** 3
**Presentation:** 2
**Significance:** 3
**Originality:** 2
**Overall Recommendation:** 4
**Confidence:** 3

**Summary:**

This paper proposes a novel neural architecture search (NAS) framework that identifies optimal neural network designs by leveraging fine-grained architectural modifications as edit-effect evidence derived from prior tasks. The method utilizes a model knowledge base to guide architecture refinement, enabling efficient adaptation across tasks.

**Compliance With Llm Reviewing Policy:**

Affirmed.

**Final Justification:**

Thanks to the author for addressing my concerns, I choose to give a positive rating.

**Key Questions For Authors:**

Please see weaknesses.

**Limitations:**

Yes

**Strengths And Weaknesses:**

Strengths:

1. Understanding how each fine-grained architectural modification affects performance across diverse tasks is a meaningful and important direction for advancing NAS research.
2. The release of a large-scale model knowledge base containing 67,760 GNN architectures represents a significant contribution to the community.

Weaknesses:

1. The proposed method appears to be general and not inherently specific to graph data. The problem formulation and motivation are also discussed from the perspective of general NAS limitations. Why, then, are the experimental evaluations conducted only on graph datasets? It would strengthen the paper to justify this design choice or to include experiments on non-graph domains.
2. How is the initial architecture determined? Additionally, how sensitive is the modification process to different initial architectures? An analysis of how varying the initialization affects the modification trajectory and final performance would improve the clarity and robustness of the study.
3. Although the authors state that all baselines are equipped with the same knowledge base, some baselines may not actually require such a knowledge base in practice. Moreover, constructing the knowledge base itself may incur substantial computational and financial costs. Could this lead to a larger effective budget for the proposed method compared to the baselines?
4. The authors should further investigate how the scale of the knowledge base affects performance. An study on different knowledge base sizes would provide more insight into the method’s scalability and robustness.

---

> ### Author Rebuttal · Authors · 2026-03-30
>
> We sincerely thank you for recognizing the value of fine-grained cross-task architectural modification and the significance of releasing a large-scale knowledge base. Below we address your questions on scope, initialization, fairness, and scalability, **some of which are already presented in the paper**.
>
> **BwDF-W1: The method appears general; why are the experiments only on graph datasets?**
>
> **Reply to BwDF-W1:** Thank you for recognizing the generality of our method. **In Tab. 4 and 5, the paper already includes experiments beyond graph datasets, including image and tabular domains.** For those experiments, we used NATS-Bench and HPOBench to replace our MKB about graph learning. These results demonstrate that our method can deliver top-tier models for non-graph data modalities.
>
> **We also explained our graph focus in Appx. B.3.** The method itself is **not graph-specific**. We instantiate it on graph learning because graph model design is especially **data-sensitive**: different datasets can favor very different aggregation, normalization, and message-passing choices. There is no universal GNN architecture that works well across all graph types. This is why the graph domain is a strong stress test for adaptive evidence weaving. We will highlight these results in our paper.
>
> **BwDF-W2: How is the initial architecture determined? How sensitive is the process to different initial architectures?**
>
> **Reply to BwDF-W2: We presented the initialization mechanism in Appx. B.3.** We first compute the prior similarity $S_0(D_u,D_i)$ by analyzing the data statistical distance between the unseen and benchmark tasks. For graph data, at $t=0$, our released code considers graph topology, such as degree, number of nodes, density, homophily ratio, clustering coefficient, betweenness centrality, feature dimensions, etc. These data statistics are passed to statistical similarity measurements, such as average Kendall's $\tau$ and PCA. After computing prior similarities, M-DESIGN constructs the initial architecture by taking a **similarity-weighted selection of the best-performing architecture choices on each benchmark. Fig. 8 shows that this initialization performs better than PCA and GraphGym-based alternatives.**
>
> **We add a new study showing the initializations is insensitive to final performance**. We use three initialization strategies that start from different initial model architectures in the MKB, with or without the similarity prior: similarity-weighted average, majority vote, and the most similar ones. The resulting **final best accuracies were close** on average, suggesting that initialization affects the starting point more than the final refined solution. We will add this analysis in our paper.
>
> **(Please refer to the same table in Reply to D48Y-Q1)**
>
> **BwDF-W3: Does the shared knowledge base imply a larger effective budget for the proposed method?**
>
> **Reply to BwDF-W3:** This is an important distinction for two lines of work. The MKB is a **shared offline resource**—conceptually closer to a benchmark, model zoo, or pretraining corpus—whereas the reported refinement budget refers to the online cost on an unseen task. M-DESIGN's advantage is precisely that it **converts prior offline investment into cheaper online refinement**, just as retrieval systems or pretrained backbones **amortize cost** across future tasks. We believe that this structured reuse provides real benefits and is a significant direction, just as skills in OpenClaw do.
>
> For the offline cost, please refer to **Reply to D48Y-Q2** for details. The one-time offline cost and the repeated online budget cannot be compared directly.
>
> **BwDF-W4: How does the scale of the knowledge base affect performance?**
>
> **Reply to BwDF-W4:** We agree and prepared a new ablation to show that **M-DESIGN is useful even before full MKB saturation**. As a sanity check on benchmark-task coverage, we ran a full ablation study across all datasets, randomly retaining only 25% / 50% / 75% / 100% of the available benchmark tasks. As shown below, **performance degraded gracefully**, with the best average accuracy moving from 82.11 at full coverage to 81.50 at 25% coverage. We will add such analysis to our paper:
>
> |MKB Size|Act|Comp|Photo|Cit|CS|Cora|Corn|DBLP|Pub|Tx|Wis|Avg|
> |---|---|---|---|---|---|---|---|---|---|---|---|---|
> |25%|34.12±1.09|88.49±1.38|94.62±0.57|74.19±2.15|95.06±0.09|88.32±0.09|74.78±3.37|84.00±0.82|**89.08±0.45**|83.79±3.82|90.00±4.32|81.50±1.65|
> |50%|**34.89±1.97**|88.45±1.49|94.62±0.57|**74.59±1.77**|95.21±0.29|88.26±0.46|75.68±4.41|**84.29±1.38**|88.70±0.38|81.08±2.20|**91.33±3.40**|81.55±1.67|
> |75%|34.28±1.00|88.49±1.38|94.62±0.57|74.19±2.15|94.89±0.10|88.38±0.66|**77.48±3.37**|**84.29±1.38**|**89.08±0.45**|**84.68±3.37**|90.00±3.27|81.85±1.61|
> |100%|**34.89±1.97**|**89.22±1.15**|**94.75±0.82**|**74.59±1.77**|**95.33±0.39**|**88.50±0.83**|**77.48±3.37**|**84.29±1.38**|**89.08±0.45**|83.79±3.82|**91.33±3.40**|**82.11±1.76**|

---

> > ### Author Rebuttal · Reviewer_BwDF · 2026-04-01
> >
> > Thanks to the author for addressing my concerns, I choose to give a positive rating.

---

> > > ### Author Response · Authors · 2026-04-01
> > >
> > > Thank you very much for your valuable feedback and for the positive update. We really appreciate your thoughtful suggestion and will incorporate those new analyses in the paper.

---

### Official Review · Reviewer_D48Y · 2026-03-12

**Soundness:** 4
**Presentation:** 4
**Significance:** 3
**Originality:** 3
**Overall Recommendation:** 5
**Confidence:** 3

**Summary:**

This paper introduces M-DESIGN, a retrieval-augmented model refinement framework designed to bridge the gap between computationally expensive Neural Architecture Search (NAS) and rigid, static model retrieval. Instead of retrieving entire model checkpoints, the proposed method treats fine-grained architectural modifications and their associated performance gains as transferable evidence. The authors structure this prior knowledge into a "modification-gain graph" within a comprehensive Model Knowledge Base (MKB).

To adaptively refine models for unseen tasks, M-DESIGN employs "knowledge weaving" to aggregate expected modification gains from similar historical tasks. To overcome the inherent limitations of static task similarity and out-of-distribution (OOD) shifts, the framework incorporates two key mechanisms: (1) a Bayesian online update that dynamically calibrates task similarity beliefs based on real-time evaluation feedback, and (2) a predictive task planner (an edge-regression GNN) to estimate unseen multi-hop gains when direct historical evidence is sparse.

**Compliance With Llm Reviewing Policy:**

Affirmed.

**Key Questions For Authors:**

Cold-start Stability: How does M-DESIGN maintain stability in the Bayesian similarity update during the very early stages of refinement (e.g., $t \le 3$), when the historical feedback required for calculating the likelihood is extremely sparse?

How responses would change evaluation: Clarifying the robustness mechanisms during this "cold-start" phase—or showing empirically that early variance does not degrade final performance—would fully resolve my primary technical concern (Weakness 1) and solidify the robustness of the method.

Offline MKB Construction Cost: Could you quantify the offline computational cost (e.g., in GPU hours) required to construct the exhaustive 1-hop Modification-Gain Graph for the benchmark datasets in your MKB?

How responses would change evaluation: Addressing this would help contextualize the scalability of the framework to larger model spaces. If the offline cost is manageable, or if there are strategies to sparsely approximate the graph without exhaustive evaluation, it would alleviate Weakness 2 and potentially elevate my view on the paper's broad significance.

Hyperparameter Sensitivity and Reproducibility: How sensitive is M-DESIGN's performance to the OOD threshold ($\tau$ or $\beta$), and what are the specific training details (e.g., epochs, data splits from the replay buffer) for the EdgeConv predictive task planner?

How responses would change evaluation: Providing a brief sensitivity analysis and committing to including these specific training details in the camera-ready appendix would fully address Weakness 3, resolving minor reproducibility concerns.

**Limitations:**

Yes

**Strengths And Weaknesses:**

Strengths:

1.Originality: The paper presents a highly original paradigm shift in automated model design. By transitioning from the conventional retrieval of static "model checkpoints" to the retrieval of "modification edges" (edit-effect evidence), the authors creatively combine the efficiency of model retrieval with the adaptability of NAS. The formulation of the Modification-Gain Graph to render historical knowledge composable is insightful and novel.

2.Soundness: The methodological framework is mathematically sound and elegantly designed to address realistic challenges. The integration of Bayes' Theorem (Eq. 8) to dynamically update task similarity effectively resolves the "static similarity" failure prevalent in prior arts. Furthermore, the theoretical assumptions—specifically the linearity of modification consistency and the Gaussian distribution of likelihoods—are rigorously validated through empirical tests (Shapiro-Wilk test and $R^2$ in Section 5.3), which significantly strengthens the paper's technical credibility. The fallback mechanism utilizing an edge-regression GNN for OOD scenarios is also pragmatic and complete.

3.Significance: The paper tackles a highly practical bottleneck in AutoML: achieving high performance under strict computational budgets. Consistently finding optimal models within just 100 evaluations is highly valuable for real-world applications. Additionally, open-sourcing the large-scale Model Knowledge Base (MKB), containing 67,760 evaluated GNN configurations with fine-grained topological and performance metadata, is a substantial contribution to the community that will facilitate future research.

4.Presentation: The paper is exceptionally well-written and structured. The narrative flows logically from identifying the dual challenges of "evolving transferability" and "OOD shifts" to systematically addressing them. Figures 1 and 3 are highly informative and clearly convey the motivation and the proposed system workflow.

Weaknesses:

Cold-start Stability of Bayesian Updates: While the Bayesian online update is elegant, its stability during the very early stages of refinement (when $t$ is small) is not fully discussed. Because the likelihood estimation relies on historical feedback, it is unclear if the similarity weights might fluctuate drastically or become noisy when only 1 or 2 real evaluations have been observed. A brief discussion on ensuring robustness during this "cold-start" phase of refinement would be beneficial.

Computational Overhead of MKB Construction: Although M-DESIGN drastically reduces the search cost for unseen tasks, constructing the exhaustive 1-hop Modification-Gain Graph for the benchmark datasets appears to require significant offline computational resources. The authors should briefly clarify the offline cost required to populate this dense graph.

Hyperparameter Sensitivity and Details: The method introduces several key thresholds and hyperparameters, such as the OOD threshold ($\beta$ in Algorithm 1, denoted as $\tau$ in Section 4.3) and the training details of the predictive task planner (EdgeConv GNN). A sensitivity analysis or a more detailed explanation of how to optimally set these parameters across diverse domains would improve reproducibility.

To further improve the camera-ready version of this manuscript, the au-thors are encouraged to address the following minor typos and formatting issues:

1.Notation Inconsistency (OOD Threshold): In Section 4.3 (Line 250), the OOD threshold is denoted as $\tau$ (e.g., $\mathcal{S}_t(D^u, D^i) \le \tau$). However, in Algorithm 1 (Lines 682), the same threshold is denoted as $\beta$. Please unify the notation.

2.Equation (5) notation: In Eq. (5), the argmax is over $\Delta\theta \in \mathcal{K}$ (Knowledge Base). However, in Eq. (7), it is over $\mathcal{C}_t$ (candidate set of feasible local modifications). Eq. (5) should probably use $\mathcal{C}_t$ since we search over feasible 1-hop edits around the current model, not the entire knowledge base.

---

> ### Author Rebuttal · Authors · 2026-03-30
>
> We sincerely thank you for the highly positive review and for recognizing the originality, soundness, practical value, and clear presentation of our work. Below we address the few points that can further strengthen robustness and reproducibility.
>
> **D48Y-Q1: Cold-start stability of Bayesian updates.**
>
> **Reply to D48Y-Q1:** M-DESIGN has three built-in stabilizers in the early stage:
>
> 1. It starts from an **explicit similarity prior** $S_0(D_u,D_i)$ by analyzing the data statistical distance between the unseen and benchmark tasks rather than an uninformative uniform belief (discussed in Appx. B.3). For graph data, at $t=0$, our released code considers graph topology, such as degree, number of nodes, density, homophily ratio, clustering coefficient, betweenness centrality, feature dimensions, etc.
> 2. The refinement loop keeps the **prior regression parameters** until there are at least 3 observed pairs for the unseen task (before that, it uses the initialized $\gamma_{i,0}=1.0$, $\sigma_{0}^2=0.01$), which prevents unstable early regressions.
> 3. The update in Eq. (9) is a **normalized Bayesian reweighting**, which smooths belief shifts instead of making hard switches.
>
> In our paper, **Fig. 7 illustrates the variations in early modification consistency**, which, in effect, reflect the expected changes in Bayesian belief. After we start to update the belief ($t>3$), the change in consistency quickly become smooths, especially after $t>20$. **Fig. 8 in Appx. B.3 shows that our initialization handles the cold-start issue better than other methods.**
>
> We add a new study showing **early variations do not significantly degrade final performance**. We use three initialization strategies that start from different initial model architectures in the MKB, with or without the similarity prior: similarity-weighted average, majority vote, and the most similar ones. The average is close:
>
> |Init Strategy|Act|Comp|Photo|Cit|CS|Cora|Corn|DBLP|Pub|Tx|Wis|Avg|
> |---|---|---|---|---|---|---|---|---|---|---|---|---|
> |Weighted Average|34.15±0.66|**89.22±1.15**|**94.75±0.82**|74.54±1.96|**95.33±0.39**|**88.50±0.83**|**77.48±3.37**|**84.29±1.38**|**89.08±0.45**|81.08±5.84|**91.33±3.40**|**81.80±1.84**|
> |Majority Vote|**34.89±1.97**|88.40±1.16|94.62±0.57|**74.59±1.77**|95.06±0.09|88.34±0.84|75.50±1.91|84.15±1.26|89.00±0.53|**83.79±3.82**|**91.33±3.40**|81.79±1.57|
> |Most Similar|33.88±2.86|88.64±1.70|94.48±1.00|74.28±1.70|94.90±0.13|88.26±0.95|74.57±2.42|84.04±1.09|88.96±0.57|81.08±5.84|**91.33±3.40**|81.31±1.97|
>
> **D48Y-Q2: Offline MKB construction cost.**
>
> **Reply to D48Y-Q2:** We agree that our paper should report this offline, one-time cost separately from our strong efficiency during online refinement. Specifically, the released MKB occupies **~47.15 MB** and contains 67,760 evaluated model configurations across 33 task-dataset DBs. Among them, each modification-gain graph occupies **~1178 KB** and each pretrained predictive task planner occupies **~112 KB**. The one-time benchmark population cost in GPU-hours depends on the device and the size of the benchmark dataset. For example, the longest data collection on our RTX 3080 GPU was *Physics*, which took **~37.59 GPU-hours** to fully populate the Modification-Gain graph, while the fastest is within **1 hour**. We will add those numbers to the camera-ready.
>
> Besides, it is not a must to fully populate the Modification-Gain graph. In case an edge is missing, the predictive task planner could use its neighborhood information to predict the modification gains, as discussed in **Sec. 4.3**.
>
> **D48Y-Q3: Hyperparameter sensitivity and EdgeConv details.**
>
> **Reply to D48Y-Q3:** In **Appx. B.3, Fig. 8 shows that our method is insensitive to $\tau$** and can always invoke the predictive task planner for better empirical results. In **Reply to BwDF-W4**, M-DESIGN is empirically good at utilizing more knowledge sources (even they are OOD) rather than relying on a carefully tuned threshold.
>
> Moreover, as detailed in **Appx. C.1**, the predictive task planners are implemented as **two-layer EdgeConv** models with hidden dimension 64. As detailed in our code, we use a 5% validation split, Adam (lr=0.01), and early stopping with patience 20 for the planner pretraining. Online adaptation is implemented by storing recent unseen-task observations in a small replay buffer and using them to incrementally fine-tune the predictive task planners. The buffer size is set to 10 and updated in FIFO fashion. This lightweight planner update supports blending buffer samples with historical edges from the benchmark graph. We will add those details to the camera-ready.
>
> **Typos issues.**
>
> We will fix both issues:
>
> - unify **$\tau$** and **$\beta$** into a single OOD-threshold notation **$\tau$**, and
> - replace the inconsistent **$K$** in Eq. (5) with the correct local 1-hop candidate set **$C_t$**.

---

### Official Review · Reviewer_qe2J · 2026-03-12

**Soundness:** 2
**Presentation:** 3
**Significance:** 2
**Originality:** 3
**Overall Recommendation:** 4
**Confidence:** 3

**Summary:**

The authors study a retrieval-augmented approach to neural network design, aiming to bridge the gap between expensive neural architecture search and static model retrieval. The core idea is to move beyond retrieving an initial model checkpoint and instead retrieve and compose fine-grained architecture-edit evidence from prior tasks. Based on this representation, the proposed method, M-DESIGN, iteratively selects edits for a new task by weaving gain evidence across related benchmark tasks with a dynamically updated Bayesian task-similarity belief. The empirical study is centered on graph learning, with a model knowledge base spanning 3 task types, 22 datasets, and 67,760 trained GNNs, and reports strong results under a fixed evaluation budget, including reaching the search-space optimum in 26 out of 33 task-data pairs.

**Compliance With Llm Reviewing Policy:**

Affirmed.

**Final Justification:**

I have read the authors' rebuttal as well as the review comments of all the other fellow reviewers. I chose to stick to my positive rating as my final justification.

**Key Questions For Authors:**

Please find the questions in the weakness section.

**Limitations:**

1.	Limited evidence of generality beyond graph learning.
2.	The full cost of constructing and maintaining the knowledge base.

**Strengths And Weaknesses:**

Strengths:

- ``S1: `` The paper shows clear problem formulation and motivation.
- ``S2: `` The paper has strong empirical results.
- ``S3: `` The model knowledge base is well spanned.

Weaknesses:

- ``W1: `` The Linearity and Gaussianity may not be convincing. For Cornell, the R square is still low.
- ``W2: `` Lack of discussing the cost of storage of 60k+ trained GNN, it would be better to discuss how to utilize them.
- ``W3: `` The efficiency metric exists the risk of favor-the-proposed-method, which is defined by M-DESIGN in 50 evaluations.
- ``W4: `` The proposed parameter exploration approach would be more convincing if it's compared against methods with fixed pre-trained Graph Neural Networks.

---

> ### Author Rebuttal · Authors · 2026-03-30
>
> We sincerely thank you for recognizing our paper's clear motivation, strong empirical results, and well-spanned model knowledge base. Below we address your concerns on modeling assumptions, evaluation fairness, and practical cost.
>
> **qe2J-W1: The linearity and Gaussianity may not be convincing. Cornell has low $R^{2}$.**
>
> **Reply to qe2J-W1:** The linearity and Gaussianity hold for many cases in our study, e.g., Cora-DBLP exhibits strong in-distribution transferability with ($R^{2}=0.87,p=0.81$), Actor-Computers with ($R^{2}=0.75,p=0.80$), etc. Cornell-Actor ($R^{2}=0.03,p=0.55$) is the intended **special case for weak-transfer & OOD stress test**, which improves after enabling the predictive task planner and OOD adaptation ($R^{2}=0.11,p=0.75$). In practice, unseen tasks may not always have a perfect match within the MKB. Such an increase in linearity validates our contribution of dynamic updates and OOD adaptation beyond static reuse.
>
> **qe2J-W2: Lack of discussing the storage cost of 60k+ trained GNN; better to discuss how to use.**
>
> **Reply to qe2J-W2:** We clarify that M-DESIGN does **NOT** require loading or serving all historical model weights during refinement. Its online reasoning can operate on a **structured knowledge schema** comprising *Tasks*, *Architectures*, and *Modification Gains* (detailed in **Appx. B.2**).
>
> Under this schema, the released MKB occupies **~47.15 MB** and contains 67,760 evaluated model configurations across 33 task-dataset DBs. For each task-dataset pair, the modification-gain graph occupies ~1178 KB, and the pretrained predictive task planner occupies ~112 KB. To utilize them, one could directly retrieve the records from the DBs or invoke pretrained GNN planners to make predictions. We will add these details to our paper.
>
> **qe2J-W3: The efficiency metric risks favoring the proposed method.**
>
> **Reply to qe2J-W3:** We introduced the thresholded efficiency metric as a convenient, unbiased fixed target for comparing how fast different methods approach our strong solution. We also have an unbiased efficiency view of the **best-so-far refinement trajectory in Fig. 4 and 9 of Appx. C.3**.
>
> We add AUC of the best-so-far refinement trajectory (higher is more efficient) below. **Our method achieves the best average AUC score:**
>
> |Method|Act|Comp|Photo|Cit|CS|Cora|Corn|DBLP|Pub|Tx|Wis|Avg|
> |---|---:|---:|---:|---:|---:|---:|---:|---:|---:|---:|---:|---:|
> |Random|0.338|0.872|0.940|0.734|0.948|0.867|0.739|0.825|0.882|0.792|0.890|0.802|
> |RL|0.336|0.873|0.941|0.736|0.948|0.874|0.738|0.825|0.884|0.800|0.888|0.804|
> |EA|0.334|0.876|0.940|0.735|0.948|0.872|0.731|0.824|0.883|0.804|0.885|0.803|
> |GraphNAS|0.337|0.871|0.941|0.734|0.948|0.872|0.735|0.828|0.883|0.807|0.888|0.804|
> |Auto-GNN|0.334|0.873|0.944|0.741|0.950|0.877|0.732|0.830|0.882|0.793|0.879|0.803|
> |AutoTransfer|0.335|0.875|0.945|0.738|0.949|**0.883**|0.750|0.833|0.887|0.780|0.880|0.805|
> |DesiGNN|0.341|0.877|0.945|0.741|0.949|0.882|0.747|**0.843**|**0.891**|**0.809**|0.898|0.811|
> |**M-DESIGN**|**0.342**|**0.885**|**0.946**|**0.743**|**0.951**|**0.883**|**0.768**|0.839|0.887|**0.809**|**0.907**|**0.815**|
>
> **qe2J-W4: Would be more convincing if compared against fixed pre-trained GNNs.**
>
> **Reply to qe2J-W4:** We add **GraphMAE** (2-layer, GAT-based, hidden size 256, mask rate 0.4, pre-trained for 200 epochs with lr $10^{-3}$, trained for 300 epochs with $10^{-2}$) as the pre-trained GNN baseline. Beyond our current focus on search-based and retrieval-based methods, our experiment shows that **M-DESIGN still consistently yields superior results**:
>
> |Method|Act|Comp|Photo|Cit|CS|Cora|Corn|DBLP|Pub|Tx|Wis|Avg|
> |---|---|---|---|---|---|---|---|---|---|---|---|---|
> |GraphMAE|26.69±0.30|75.77±4.55|88.71±4.87|72.98±0.57|94.27±0.17|87.08±1.15|45.95±7.15|84.18±1.26|85.36±0.95|54.05±7.15|50.67±11.02|69.61±3.56|
> |M-DESIGN|**34.89±1.97**|**89.22±1.15**|**94.75±0.82**|**74.59±1.77**|**95.33±0.39**|**88.50±0.83**|**77.48±3.37**|**84.29±1.38**|**89.08±0.45**|**83.79±3.82**|**91.33±3.40**|**82.11±1.76**|
>
> **qe2J-L1: Limited evidence of generality beyond graph.**
>
> **Reply to qe2J-L1: In Tab. 4 and 5, the paper includes experiments beyond graph learning, including image and tabular domains.** For those experiments, we used NATS-Bench and HPOBench to replace our MKB about graph learning. These results demonstrate that our method exhibits non-graph generality.
>
> Running our main experiments on graph learning was a deliberate choice because graph model design is especially **data-sensitive** and does not admit a single universal GNN architecture, making it a strong stress test for adaptive refinement. Appx. B.3 explains this motivation explicitly.
>
> **qe2J-L2: The cost of MKB.**
>
> **Reply to qe2J-L2: The MKB is a shared offline resource that converts prior offline investment into cheaper online refinement**, just as retrieval systems or pretrained backbones **amortize cost** across future tasks. Please refer to **Reply to D48Y-Q2** for cost details.

---

> > ### Author Rebuttal · Reviewer_qe2J · 2026-04-08
> >
> > I appreciate the authors' detailed responses and the additional experimental results. After going through the comments of other reviewers, I choose to keep my positive rating.

---

### Decision · Program_Chairs · 2026-04-30

**Decision:**

Accept (regular)

**Comment:**

The paper presents M-DESIGN, a retrieval-augmented model refinement framework that leverages "edit-effect evidence" from historical tasks to navigate neural network design. The approach is technically sound and addresses the efficiency-quality trade-off in architectural search by using adaptive retrieval and predictive task planners to handle out-of-distribution shifts.
The reviewers consistently praised the extensive experimental validation, which encompasses over 67,000 graph neural networks across 22 datasets, demonstrating superior performance against established baselines in the majority of cases. During the rebuttal and discussion period, the authors were highly engaged and successfully addressed the reviewers' primary concerns. Specifically, Reviewer 1Txj noted that their concerns (including those labeled Q1 and Q3) were fully resolved and recommended including these points in the final manuscript as a discussion on limitations. Reviewer BwDF, who initially expressed reservations, increased their score to a Weak Accept after reviewing the authors' responses.
Overall, there is a clear consensus among the four reviewers (one Accept, three Weak Accepts) that the paper provides a significant and well-supported contribution to the ICML community. The AC agrees with the reviewers that the framework's ability to "weave" historical knowledge for fine-grained design is both novel and effective.